# Task Ambiguity in Humans and Language Models

**Alex Tamkin**,[*] **Kunal Handa**[*]**, Avash Shrestha, Noah Goodman**
Stanford University

## Abstract

Language models have recently achieved strong performance across a wide range of NLP benchmarks. However, unlike benchmarks, real world tasks are often poorly specified, and agents must deduce the user's intended behavior from a combination of context, instructions, and examples. We investigate how both humans and models behave in the face of such task ambiguity by proposing AmbiBench, a new benchmark of six ambiguously-specified classification tasks. We evaluate humans and models on AmbiBench by seeing how well they identify the intended task using 1) instructions with varying degrees of ambiguity, and 2) different numbers of labeled examples. We find that the combination of model scaling (to 175B parameters) and training with human feedback data enables models to approach or exceed the accuracy of human participants across tasks, but that either one alone is not sufficient. In addition, we show how to dramatically improve the accuracy of language models trained without large-scale human feedback training by finetuning on a small number of ambiguous in-context examples, providing a promising direction for teaching models to generalize well in the face of ambiguity.

## 1 Introduction

Language models have recently been applied to a wide range of NLP benchmarks, ranging from question answering, summarization, and logical reasoning, to solving riddles, dark humor detection, and ASCII word recognition (Brown et al., 2020; Srivastava et al., 2022). Performance across tasks has improved as models and datasets have grown in size, raising the prospect of a route towards generalist NLP models with broad utility.

However, one feature many of these benchmarks share is that they are carefully designed to make the desired task very clear to the language model, since this is a prerequisite for establishing performance on that task. Unfortunately, real-world uses of language models are not likely to feature such

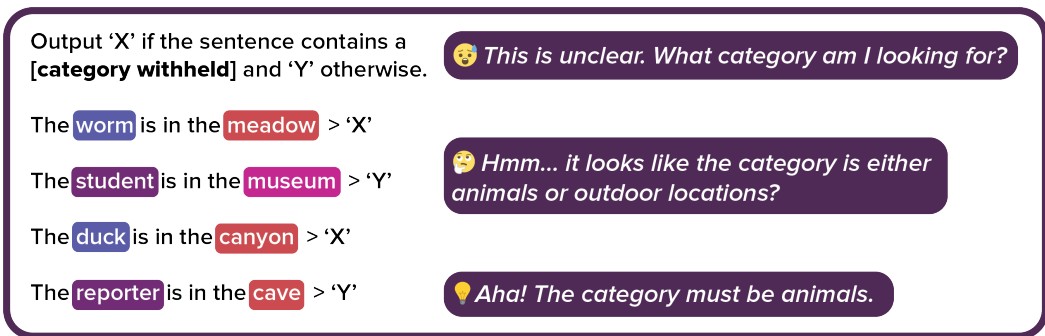

Figure 1: Complex tasks are often hard to specify precisely, leaving important pieces of information missing. Agents should be able to fill in the blanks by combining information from instructions and examples in order to identify the intended behavior.

---

[*]Equal Contribution. Correspondence to atamkin@cs.stanford.edu

thought and clarity in their task specification. Rather than iterating over and perfecting a specification for their tasks, everyday users of language models may wish to define tasks on an as-needed basis, without worrying that they will be misunderstood. More pressingly, in complex domains featuring high-dimensional inputs and outputs (e.g. programming, verification, generation) it is unlikely that even a thoughtful task specification will manage to perfectly capture all the features of an input and output which are salient or not salient to the task. This is especially important for safe and robust deployment of language models, as such undesirable dependencies can be hidden hazards that are only revealed when a model fails catastrophically in a new setting (Geirhos et al., 2020).

To operationalize this problem, we introduce AmbiBench, a new benchmark of six ambiguously-specified tasks. Each input in AmbiBench is a sentence (e.g. *The dog is in the meadow*) that has multiple associated classification tasks based on different linguistic features (e.g. *contains an animal*, *contains an outdoor location*). Task ambiguity arises when more than one task is consistent with the provided instructions or labeled examples.[1]

We establish how well different models and humans perform on ambiguously-specified tasks, given a wide range of task specifications including clear vs unclear instructions and zero vs multiple examples. We find that the largest models trained with human feedback data (HFD) match or outperform human participants across all specifications we try, though all underperform a Bayesian oracle.

We also show how to improve standard language models' performance by finetuning them on a small set of in context examples that demonstrate the desired generalization. This form of meta-learning dramatically improves a model's ability to learn new ambiguously-specified tasks. This suggests a possible mechanism for why the HFD models outperform standard language models (discussed in Section 4.4), as well as a promising direction for improving how models learn in ambiguous contexts.

To summarize our contributions, we:

1. Introduce and motivate the problem of studying task ambiguity in large language models

2. Evaluate humans and models on a new benchmark of ambiguously-specified tasks, demonstrating that while pure language models fail to disambiguate the intended task well, sufficiently-large models trained with human feedback data are able to approach or even exceed the performance of our human participants to resolve the ambiguity between tasks

3. Show how finetuning on ambiguous in-context prompts and examples can enable traditional language models to surpass the performance of HFD models when evaluated on unseen tasks, providing a promising route towards models that capably manage task ambiguity

## 2 RELATED WORK

### 2.1 AMBIGUITY IN NATURAL LANGUAGE PROCESSING

Ambiguity is a well-studied topic in NLP, with work spanning topics as diverse as search queries (Cronen-Townsend & Croft, 2002; Wang & Agichtein, 2010), question answering (Min et al., 2020; Zhang & Choi, 2021), named entities (Bunescu & Pasca, 2006; Cucerzan, 2007; Dredze et al., 2010), coreference resolution (Webster et al., 2018), machine translation (Stanovsky et al., 2019), and information-seeking dialogues (Aliannejadi et al., 2019; Guo et al., 2021; Aliannejadi et al., 2021; Sun et al., 2022; Wu et al., 2022).

Our work differs from these prior streams of work by studying *task ambiguity* (Finn et al., 2018; Tamkin et al., 2022c), where the task the agent is being asked to perform is ambiguous, rather than an ambiguous input for a clear task. This is of special relevance for self-supervised learning models that are trained for adaptation to a broad range of downstream tasks (Bommasani et al., 2021; Tamkin et al., 2022b). In these settings, models must infer the correct task from a user's specification, as opposed to a possibly unsafe or undesirable task that is also consistent with that specification.

---

[1]Importantly, task ambiguity is distinct from clearly-specified tasks with *ambiguous inputs*, e.g. determining the referent of the pronoun in sentences like *the nurse handed the doctor **her** phone*. Here, the task is clear (determine who **her** refers to), but there is not enough information in the input to answer it.

## 2.2 IN-CONTEXT LEARNING AND PROMPTING

Task ambiguity is especially relevant for language models, which can be adapted for many different tasks via in-context learning (Brown et al., 2020; Tamkin et al., 2021a; Bommasani et al., 2021; Liu et al., 2022b), and may rely on undesirable different linguistic features to solve a task (Gururangan et al., 2018; Tamkin et al., 2020). Much work has attempted to improve the ability of such models to perform in-context learning by calibrating model predictions (Zhao et al., 2021), choosing good examples for the prompt (Liu et al., 2022a), finetuning models on natural language descriptions of tasks (Zhong et al., 2021; Wei et al., 2022; Sanh et al., 2022), or by training models with reinforcement learning from human feedback (Bai et al., 2022; Ouyang et al., 2022).

Prior work has suggested that language models may not effectively learn from the provided instructions (Webson & Pavlick, 2022) or few-shot examples (Min et al., 2022b; Kim et al., 2022); instead such models may rely on cues such as the formatting of the examples or the label space. In this work, we present a way to measure how well models use instructions or few-shot examples that is unaffected by such cues, because each AmbiBench example is consistent with *multiple* possible tasks. Thus, models that perform well on AmbiBench must infer the desired task using e.g., the task instruction or other examples. This enables a clean empirical investigation of how well large language models serve as Bayesian reasoners, as past work has hypothesized (Xie et al., 2021).

Past work has also explored finetuning on in-context learning examples (Chen et al., 2022; Min et al., 2022a). We extend this line of work to show how the content of these training examples can dramatically affect generalization: Finetuning on *ambiguously-specified* examples (but not a control set of unambiguous tasks) can enable models to disambiguate better in new settings—vastly improving the performance of pure language models without the need for human feedback data.

## 2.3 TASK AMBIGUITY

Systems capable of performing different tasks may experience *task ambiguity*, where the provided examples do not uniquely identify the user's intended task (Finn et al., 2018; Tamkin et al., 2021a). One form of task ambiguity is shortcut learning (Geirhos et al., 2020), where the training examples can all be solved by identifying a simple feature (e.g. a watermark) as opposed to learning the intended task (e.g. object classification). Task ambiguity is particularly important in few-shot learning settings, where the small number of examples may leave the intended task ambiguous (Finn et al., 2018; Tamkin et al., 2021a). In this work, we study task ambiguity for in-context learning of simple linguistic tasks, considering not only the role of examples but also natural language instructions.

## 3 THE AMBIBENCH BENCHMARK

As a first step towards studying task ambiguity in language models, we construct the AmbiBench benchmark, a collection of six different sentence classification tasks. The goal of AmbiBench is to construct a testbed of *minimal complexity* where we can control and measure the degree of ambiguity in various task specifications. Despite the simplicity of this benchmark, we find large variability in performance across different language models.

### 3.1 SELECTION OF TASKS

AmbiBench contains six binary classification tasks, where a human or model must detect a simple linguistic feature in an input sentence—for example, whether an outdoor location or an animal was mentioned—and then *output* the appropriate classification letter (X or Y). Crucially, however, each sentence has two linguistic features (e.g. *The duck is in the canyon* has the features *animal* and *outdoor location*). The six features are grouped into three pairs, shown in Table 1, where a single sentence will have one feature in each pair.

To identify the *salient feature* for the task, then, one must have either an informative instruction (e.g., *Output 'X' if the sentence contains an outdoor location and 'Y' otherwise*) or multiple labeled examples to disambiguate which feature determines the label.

Tasks were chosen to represent a set of common semantic categories, excluding subtoken information such as periods and capitalization that might be much easier for humans to represent than

| Salient feature | Example sentence |
|---|---|
| human subject | The **researcher/bear** is in the museum. |
| indoor location | The researcher is in the **museum/meadow**. |
| religious leader | He is in the museum with the **rabbi/judge**. |
| pronoun gender | **He/She** is in the museum with the judge. |
| proper noun | **Paul Atreides/The director** may not be in the film studio. |
| negation | Paul Atreides **may/may not** be in the film studio. |

| Instruction | Example |
|---|---|
| Uninformative | *Output 'X' if the sentence contains a **[category withheld]** and 'Y' otherwise.* |
| Informative | *Output 'X' if the sentence contains a **proper noun** and 'Y' otherwise.* |

Table 1: **The AmbiBench benchmark**. *Left*: Each task involves detecting a salient feature in a sentence (bolded in the examples on the right). The same sentence could potentially receive a label according to two features, requiring a learner to use additional information (task instructions or other examples) to disambiguate the intended behavior. *Right:* Varying levels of instruction are inserted before the examples, providing different degrees of information about the format and salient feature of the task. See Figure 1 for an example of a complete prompt.

models. See Figure 1 for an example of this disambiguation process, and Table 1 for a full list of tasks and accompanying instructions.

## 3.2 TASK CONSTRUCTION

AmbiBench examples are programmatically constructed from a set of templates, allowing precise control over the amount of task ambiguity in each in-context example (see Table 1 and Appendix G for more details). Templated data has seen a recent resurgence in NLP for the purposes of evaluating large language models (Lake & Baroni, 2018; Srivastava et al., 2022), as they enable precise control and coverage over different variables of study. Furthermore, recent work has shown strong correlation between test performance on synthetic and naturalistic data Liu et al. (2021), suggesting that insights gained from such datasets may extend to a broader range of natural contexts. In our case, this dataset construction process enables us to formalize and characterize the degree of task ambiguity in different examples, allowing us to measure how well models can disambiguate between multiple potential classification tasks they may be asked to perform.

## 3.3 IN-CONTEXT LEARNING FORMATS

There are several ways an instruction and in-context examples can be assembled into a prompt for a language model. Given the demonstrated sensitivity of models to such parameters (Zhao et al., 2021; Liu et al., 2022b; Lu et al., 2022), we consider two different prompt formats, and report averaged performance across them:

**Arrow:**
```
Output 'X' if the sentence
contains an outdoor location
and 'Y' otherwise.
The worm is in the meadow
>X
The duck is in the canyon
>Y
...
```

**Q/A:**
```
Output 'X' if the sentence
contains an outdoor location
and 'Y' otherwise.
Q: The worm is in the meadow
A: X
Q: The duck is in the canyon
A: Y
...
```

## 4 EXPERIMENTS

We use AmbiBench to investigate how humans and language models respond to and resolve different manifestations of task ambiguity.

### 4.1 EXPERIMENTAL SETUP

First, we describe the different language models and human participants we study, and how we evaluate them.

**Language models**  We examine a range of different models, including both OpenAI's normal language models and their "instruct" models trained with human feedback data (HFD) (Brown et al., 2020; Ouyang et al., 2022). These models are trained using the data described in Ouyang et al. (2022) as well as on highly-rated model generations.[2]  In the rest of the paper, OpenAI's model names are reported as listed in their documentation [3] (e.g. `davinci`, `text-curie-001`). The instruct models have a numerical suffix (e.g. `002`) and the model size increases as one progresses through the alphabet (`ada`, `babbage`, `curie`, `davinci`).  See Appendix E for more information.  We also evaluate AI21 Studio's 178B-parameter Jurassic-1 Jumbo language model (`jurrasic-jumbo`) (Lieber et al.), as well as the 11B-parameter T0++ (`t0pp`) (Sanh et al., 2022) and Flan-T5 (`flan-t5`) models (Chung et al., 2022), which were finetuned on a large corpus of task instructions. This diversity of model providers, model sizes, and training strategies enables us to identify which ingredients are most crucial for resolving task ambiguity.

**Human evaluation**  We compare model performance with the performance of human participants, evaluated by hiring contractors from Prolific (Palan & Schitter, 2017). We aimed to evaluate the human participants as similarly to language models as possible within the confines of an online survey methodology.  We showed human participants exactly the same input that language models received, with minimal additional information presented to them before the study began. Participants typed the answer label (i.e. `X` or `Y`) into a textbox, as opposed to choosing from a set of preselected options, to mitigate priming effects and mirror the setting for language models. We also recruited a new participant for every single in-context instance, to avoid humans learning across examples in ways that language models do not.  Human participants were paid $12-13/hr, in line with Prolific wage recommendations.[4]. See Appendix F for more details.

### 4.2 TASK DISAMBIGUATION USING NATURAL LANGUAGE INSTRUCTIONS

One way that people resolve task ambiguity is through the use of natural language instructions, which can explicitly indicate different aspects of the task. Past work has suggested that the best models do not fruitfully use natural-language instructions, as evidenced by experiments leveraging irrelevant or misleading directions (Webson & Pavlick, 2022). However, these experiments were performed for established natural language processing tasks that lack the explicit task ambiguity we study here, and did not investigate more recent models trained with human feedback data (Bai et al., 2022; Ouyang et al., 2022).

As a first set of experiments, we evaluate how humans and models are able to use differing levels of instruction to resolve task ambiguity. The humans and models receive two in-context examples, one from each class. Humans and models are then presented with a third query example in order to elicit the predicted output letter. Because there is only one example of each class, but two possible features, the salient feature can not be identified from these two examples alone, requiring the model to use the instruction to disambiguate the task. The order of the examples, the example format, as well as the assignment of each class to an output letter (`X` or `Y`) are randomized.  Each model is evaluated with 720 different in-context prompts for each level of instruction.

We consider two different levels of instruction:

---

[2]https://beta.openai.com/docs/model-index-for-researchers
[3]https://beta.openai.com/docs/
[4]https://www.prolific.co/pricing

1. **Informative instruction**: The model receives a full specification of the salient feature and output format. Ex: *Output 'X' if the sentence contains an animal and 'Y' otherwise.*

2. **Uninformative instruction**: The model receives the output format but the salient feature is redacted. Ex: *Output 'X' if the sentence contains a [category withheld] and 'Y' otherwise.*

Our setting is simple enough that crafting an informative instruction is not challenging, making it tractable for us to study. However, the insights from this simple case may generalize to more complex settings where users may be prone to accidentally omit crucial information from the prompt.

### 4.2.1 RESULTS

In the case of **uninformative instructions**, humans as well as many models are able to achieve approximately 50% accuracy by correctly understanding the output format and choosing X and Y at random. However, some non-instruct models, including `jurassic-jumbo`, `ada`, `babbage`, and `curie`, often output values other than X or Y (e.g. Z), leading to lower performance. Finally, in the case of negation, humans achieve 100% accuracy despite lacking an instruction identifying the salient feature. This may be due to an inductive bias present in people (but not models) that makes negation an especially salient feature.

In the case of **informative instructions**, humans perform the strongest at this task, with perfect performance in all but one task, showing that they are broadly able to identify the salient feature in the text inputs and output the correct letter. Humans are closely followed by the `text-davinci-003` and `text-davinci-002` HFD models (see Figure 2). All other models perform relatively poorly, including the non-HFD 175B+ parameter `davinci` and `j1-jumbo` models, as well as the smaller HFD models `curie`, `babbage`, and `ada` (although we verify in Section that these models are still able to generate outputs in the correct format). This seems to suggest that most models are not reliably able to follow simple instructions to disambiguate a task, but that a *combination* of large-scale training and HFD can approach human performance in some settings.

### 4.3 TASK DISAMBIGUATION USING MULTIPLE EXAMPLES

While instructions are a simple way to specify a task, multiple examples can also disambiguate between different tasks a user might intend. For example, if there are multiple features that could explain the label of a single example, more examples will gradually identify the salient feature provided the features are sufficiently decorrelated.

We investigate whether models and humans can identify the salient features in AmbiBench as the number of examples grows from zero (where the task is completely ambiguous) to twenty (where the task is almost certainly unambiguous). Both models and human participants predict the answer for each example, then are presented with the correct answer for that example and the next query.[5] All aspects of the examples are randomized, including the salient feature (chosen randomly, then held constant across the entire in-context example), the assignment of X or Y to the salient feature, the example order, and the specific instantiations of the salient and non-salient features for each example. Each model is evaluated with 720 different in-context prompts, each containing 20 examples.

We also compare humans and models with a Bayesian oracle that represents how well an optimal learner could perform on the benchmark. This oracle performs perfectly as soon as it sees a set of examples which disambiguate the intended task, and performs at chance otherwise.

### 4.3.1 RESULTS

To our surprise, the best language model (the HFD-trained `text-davinci-002`) significantly outperformed the human participants (Figure 3). The human participants performed comparably to the `j1-jumbo` and `curie` models, which in turn performed better than the rest of OpenAI's models. The `flan-t5` and `t0pp` models performed worse, with the latter exhibiting large sensitivity to the prompt format—`t0pp` outputted invalid answers (typically nouns) for the **arrow** format. However, considering only the **Q/A** format, `t0pp` still only performed near chance. All models

---

[5]For causal language models, such as OpenAI's models and J1-Jumbo, which only attend backwards, this can be done efficiently by presenting the full 20 examples to the model and looking at the probability assigned to the correct answer for each example.

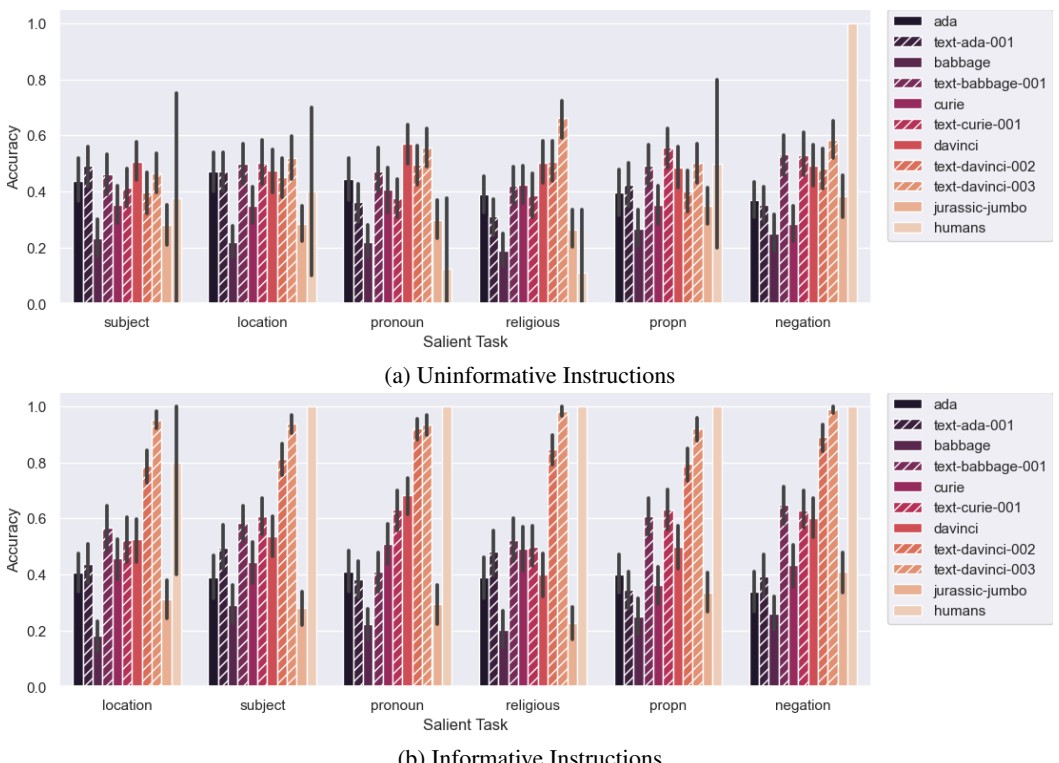

(a) Uninformative Instructions

(b) Informative Instructions

Figure 2: **The best HFD model (`text-davinci-003`) approaches human accuracy for both uninformative and informative instructions.** Accuracy of humans and other models for tasks prompted with an instruction and two in-context examples. Error bars show 95% bootstrap CIs.

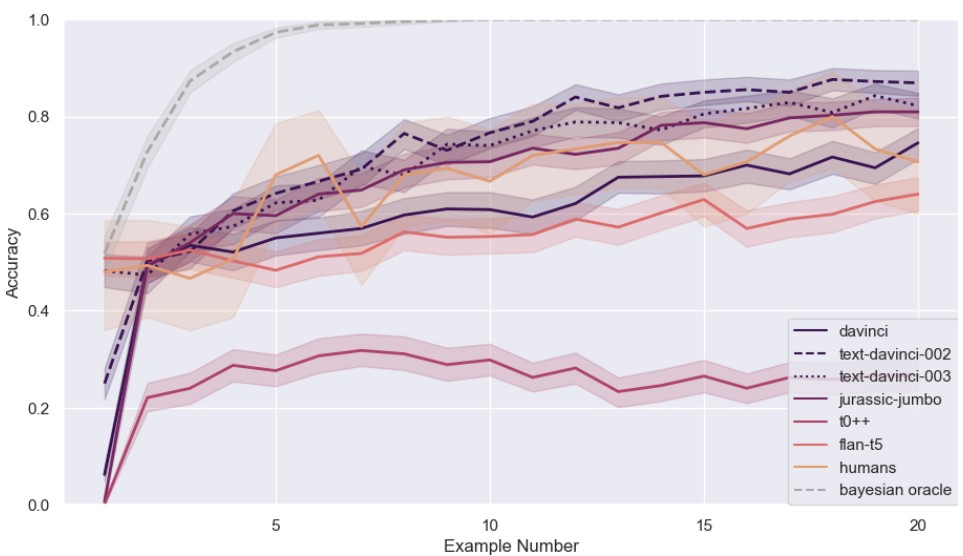

Figure 3: **The best HFD models (`text-davinci-002` and `text-davinci-003`) outperform human participants at disambiguating the intended task.** Accuracy as the number of examples in the in-context window grows. Surprisingly, the smaller `curie` model reliably outperforms the larger `davinci` model across the examples. In addition, the HFD training hurts at `curie` scale, but dramatically helps at `davinci` scale. Shaded regions are 95% bootstrap CIs.

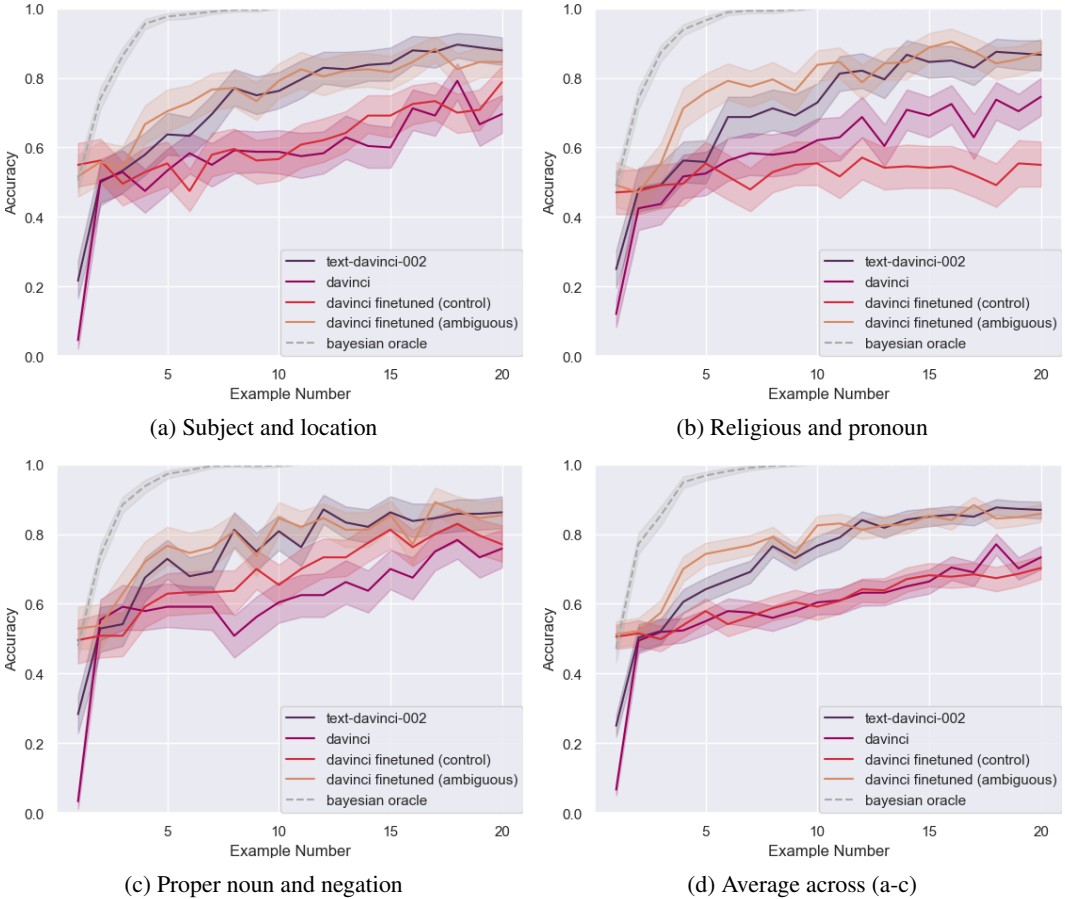

(a) Subject and location

(b) Religious and pronoun

(c) Proper noun and negation

(d) Average across (a-c)

Figure 4: **Finetuning on ambiguous in-context examples dramatically improves accuracy on unseen tasks that are ambiguously specified**. Accuracy after finetuning `davinci` on ambiguous and non-ambiguous (control) in-context examples. Models are finetuned on 272 examples from four tasks, then evaluated on the two held-out tasks (subfigure captions). Shaded regions are 95% bootstrap CIs.

considerably underperform the Bayesian oracle, suggesting room for additional improvement. See Appendix A for initial experiments investigating whether models can describe the few-shot task in words.

We do not observe evidence that the imperfect human performance is due to low-quality participants or bot activity—human annotators mostly spent between 4 and 8 minutes on the task, did not appear to be guessing at random, and typically left thoughtful comments or feedback on the survey. That said, we caution against claims of "superhuman performance" given that annotators represent merely a sample from a single distribution of humans, and they may have experienced fatigue or distraction across the 20-example episode.

### 4.4 FINETUNING A MODEL TO GENERALIZE WELL IN THE FACE OF AMBIGUITY

The strong performance of the HFD models relative to the normal language models in Section 4.3 is somewhat surprising—these models are described as being trained to follow human instructions, not to resolve ambiguity in instructions by analyzing the training examples. While the training dataset of these models was not released, Ouyang et al. (2022) do report that some of the crowdsourced examples for the model contain instructions along with few-shot examples. If some of these instructions were ambiguous, the model may have learned from those examples to resolve that ambiguity more effectively.

Motivated by this hypothesis, we investigate whether finetuning on a small corpus of ambiguous in-context learning examples is sufficient to close the gap between the best-performing `text-davinci-002` HFD model and the normal `davinci` language model. To do so, we partition the six AmbiBench tasks into three folds, each containing four finetuning tasks and two evaluation tasks (following the feature pairs in Table 1). We finetune on 68 examples from each task (two for each number of examples, from 4 to 20), and evaluate on 240 examples randomly drawn from the other two tasks. While all tasks share some structural similarities, this partitioning ensures that the model is being tested on held-out features that never appeared in its finetuning dataset. Models are finetuned using the OpenAI API (see Appendix H for details).

To see whether ambiguous data is the key factor when finetuning, we also finetune on unambiguous versions of this data, where only one feature varies within each in-context example. For example, if the two features are animal and indoor location, a given in-context example may contain examples with both animals and humans, but only indoor locations. See Appendix H for more details.

### 4.4.1 RESULTS

Despite the small training dataset consisting of only 4 tasks (with 272 examples total), we find we are able to completely close the gap between the HFD models and our finetuned models across all three splits of our data. Indeed, our finetuned models appear to even outperform `text-davinci-002` across the first eight examples, closing part of the gap to the Bayesian oracle.

Crucially, we do not observe any improvement for the control finetuned models, which were finetuned on the same kinds of examples but without task ambiguity between two potential salient features. This indicates that ambiguity is the crucial ingredient explaining the success of our finetuned models, and supports the hypothesis that the few-shot examples in `text-davinci-002`'s human feedback data may contribute to its strong performance.

More broadly, these results suggests that explicitly finetuning models to adapt to task ambiguity may result in a generalized capacity to do so across different kinds of ambiguous task specifications.

## 5 DISCUSSION AND CONCLUSION

We present the AmbiBench testbed for studying task ambiguity in language models and humans, showing how it can be used to investigate different factors influencing task ambiguity, as well as identify promising interventions that can improve how models resolve it.

### 5.1 LIMITATIONS

Our study has several limitations. First, we conduct a scientific and controlled study of task ambiguity in language models; this naturally elides many of the messy nuances of task ambiguity in the real world, and should be seen as complementary to in-the-wild case studies. We explore one such real-world use case in Appendix B, however more work is needed. Second, despite our efforts to match the experimental conditions between humans and language models, humans do require some additional instructions to orient them to the task interface, and may suffer from fatigue and uneven concentration across the length of a 20-example learning episode. Finally, our work studies task ambiguity between two possible tasks—however, in general task ambiguity may occur between arbitrarily many tasks, or even an infinitely large *family* of tasks.

### 5.2 FUTURE WORK

Task ambiguity is a pressing problem in machine learning with relevance for safety, fairness, and interpretability. Going forward, we are excited by the potential to study task ambiguity in self-supervised models trained on many different modalities (Reed et al., 2022; Tamkin et al., 2021b; 2022a; Alayrac et al., 2022), including multimodal settings, as self-supervised learning is applied increasingly broadly. The strong performance of models on the AmbiBench testbed also suggests the tractability of studying task ambiguity in more complex real-world settings where language models are used, such as software engineering, law, and education, as well as assessing the efficacy of our proposed finetuning interventions.

**Acknowledgments and Funding Disclosure** We'd like to thank Ben Prystawski, Shyamal Buch, and Rose Wang for useful conversations and feedback, as well as all the participants who took our study. We thank the Center for Research on Foundation Models (CRFM) and Together Computer for facilitating access to models. AT is supported by an Open Phil AI Fellowship.

**Ethics statement** Our research makes use of human subject experiments via the Prolific platform (Palan & Schitter, 2017). We pay workers a minimum of $12-13 / hour, consistent with Prolific wage recommendations.[6] We also made efforts to solicit feedback from participants via pilot studies, which led to several changes to the research methodology to make the survey experience more pleasant (e.g. keyboard shortcuts to navigate the study more efficiently). Anecdotally, many participants expressed that they enjoyed the study:

1. *Zap! I'm suddenly back in high school with Ms. Langston's English class. Thank you for the smiles!*

2. *this was fun!*

3. *this was a really good study*

**Reproducibility statement** We include detailed experimental settings in Sections 1, 4.2, 4.3, 4, E, H, G. We also release our codebase, including the benchmark data, at:
`https://github.com/kunhanda/task_ambiguity`

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

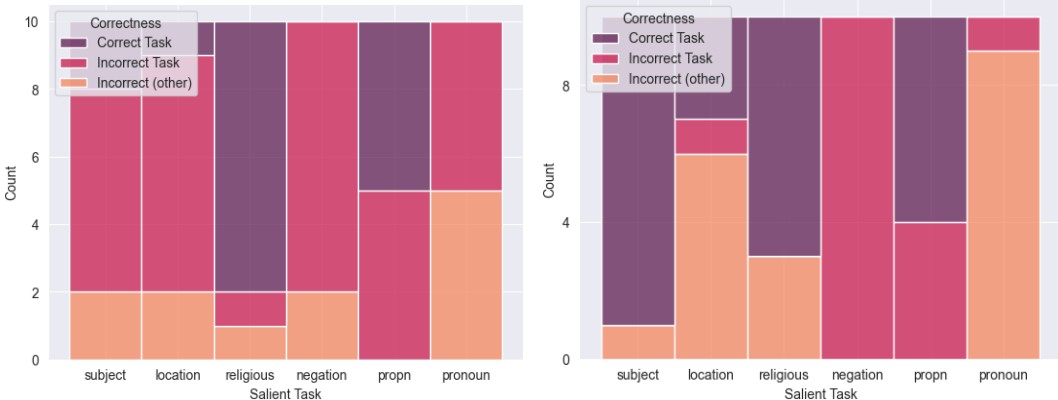

(a) Accuracy of `text-davinci-002` for guessing the [category withheld] for each salient task. Q/A format only (N=10).

(b) Accuracy of `text-davinci-003` for guessing the [category withheld] for each salient task. Arrow and Q/A formats (N=20).

Figure 5: **Even models that perform very well on a task struggle to describe that task in words.**

# A  CAN MODELS VERBALIZE THE TASK THEY ARE PERFORMING?

As models attempt to disambiguate between different tasks, a user may wish to know the model's best estimate of the task it is being asked to perform. In this section, we assess whether models can verbalize the task descriptions in Section 4.3 at the end of the in-context prompt.

Specifically, after the 20 example prompt we append a newline with the strings

- "What is the [category withheld]?" for `text-davinci-002`, and

- "What is your best guess for the [category witheld] above?" for `text-davinci-003`.

We developed the second prompt as the newer `text-davinci-003` model would often return responses along the lines of "This question cannot be answered without further context" for the first prompt.

We generate 10 outputs for each of the 6 tasks, and manually categorize them as one of three categories:

- **Correct task:** A correct verbalization of the task (e.g. "The [category withheld] is a religious leader."

- **Incorrect Task** The model guesses a task, but it is not the correct task. (e.g. "animals" when the salient task is "outdoor location")

- **Incorrect (Other)**: The verbalization does not mention a task (e.g. "The category is withheld.")

As shown in Figure 5, the models are sometimes able to verbalize the correct task, especially for religious figures and proper nouns, although for all other tasks it struggles. These initial experiments suggest that task verbalization is challenging even for the most capable models, even when they perform very well at the task, and suggests an exciting direction for future study.

Note that our graph for `text-davinci-002` exclusively considers the Q/A format, as we did not observe in a preliminary investigation that the model could verbalize the task successfully with the arrow format. Additionally, in preliminary investigations we did not find that other models were able to successfully verbalize the task.

## B EXPLORING TASK AMBIGUITY FOR A NATURAL LANGUAGE COMMAND-LINE ASSISTANT

In this section we explore how the methodologies we introduce in AmbiBench can be extended to more real-world settings. Here, we consider the case of a natural language assistant for command-line tasks. In our hypothetical setting, an employee of a company prompts a language model to produce Amazon Web Services buckets for different users by providing it with two input-output examples of the desired behavior. By chance, the two training examples both use Japanese names and have the bucket location set to the `ap-northeast-1` region in Japan. The test examples ask for a bucket to be made for a person with a non-Japanese name (in our case, White American, Greek, or Indian names).

Concretely, examples look like the following:

```
Write the AWS CLI command to create an AWS Bucket.

Input: Create a bucket for Sato Tamotsu
Output: aws s3 mb s3://bucket-for-sato --region ap-northeast-1

Input: Create a bucket for Yuki Hashimoto
Output: aws s3 mb s3://bucket-for-yuki --region ap-northeast-1

Input: Create a bucket for Margaret Richards
Output:
```

The task ambiguity that arises is that sometimes the model assumes the bucket should be placed in the same region as the training examples, but in other cases it assumes the bucket should be placed in a different region based on the person's name. We omit other commandline arguments for readability, however we note that this ambiguity might be very difficult to notice in practice with long commands consisting of many arguments.

In Figure 6 we show that not only does this ambiguity manifest in the `text-davinci-002` language model outputs (which can output both regions), but it also varies based on the national origin of the person's name. White American names induce the model to output other regions (typically ones in the US and Europe) far more than Indian or Greek names do. However, after one additional name from that same national origin is added (indicating that the name should not determine the region) the model performs nearly perfectly at other names of this national origin.

This initial investigation illustrates how task ambiguity can manifest in a more real-world setting, and how we can measure it using the techniques we discuss in the rest of the paper. Future work could explore finetuning on similar in-context examples to meta-train the model such that it avoids assuming that an individual's name should impact the task behavior, for example.

### B.1 LIST OF NAMES

Here we provide the full list of names we used to query the model. Names were constructed from a list of popular first and last names.[7]

**Japanese names**: Sato Tamotsu, Yuki Hashimoto, Yamamoto Mitsuo, Kuroda Kumiko, Mita Naoki, Tanaka Sakura, Kuramoto Hideki, Sazama Miyuki, Hora Izanagi, Itoh Akane

**White American names**: Trevor Fitzpatrick, Jessica Price, Logan Evans, Leah Mills, Ava Smith, Cole Jones, Isabella Brown, Emily Davis, Mia Miller, Madison Wilson

**Indian names**: Rajesh Kumar, Abhishek Yadav, Rahul Singh, Archana Kumar, Anisha Patel, Priya Sharma, Riya Gupta, Ayush Jain, Prisha Rao, Sagar Goyal

**Greek Names**: Yiannis Papadopoulou, Vassilis Karagiannis, Athanasios Ioannou, Dimitra Papadakis, Vasiliki Georgiou, Eleni Oikonomou, Kostas Giannopoulos, Nikos Theodorou, Ioannis Vlachos, Katerina Makris

---

[7]forebears.io

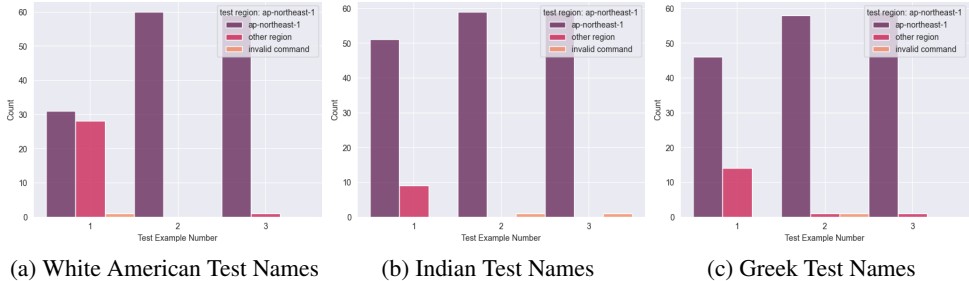

| (a) White American Test Names | (b) Indian Test Names | (c) Greek Test Names |

Figure 6: **Features such as name origin can influence how models behave under task ambiguity.** Effect of task ambiguity on `text-davinci-002` when generating an AWS bucket command given an ambiguous natural language string. Graph shows distribution of generated regions: ap-northeast-01 (the region in the prompts), another region, or invalid command.

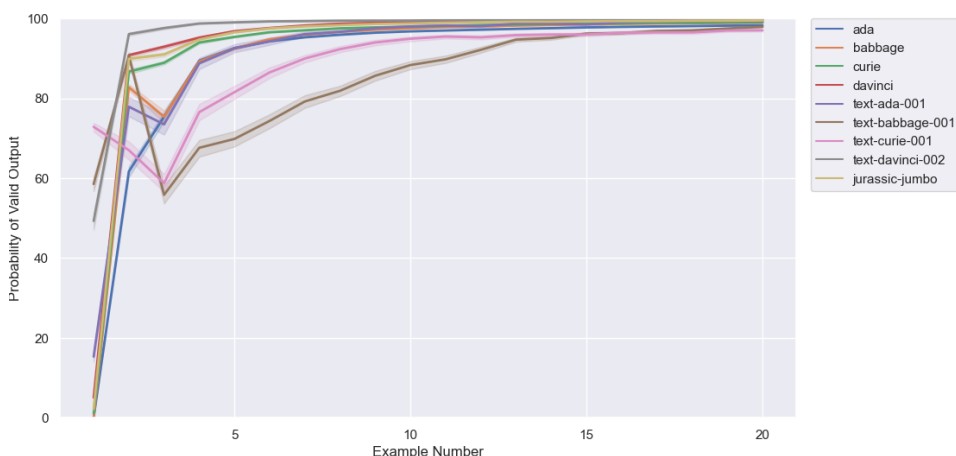

Figure 7: **Models of all sizes generate valid outputs by the end of the in-context learning episode.** Probability of a valid output (either X or Y) for all models across examples

## C   DO SMALLER MODELS UNDERSTAND THE TASK FORMAT?

In Section 4.3 we find that smaller models perform worse at disambiguating the intended task from multiple examples. However, is this due simply to models not understanding the desired output format? We validate the output formats of all models and find (Figure 7) that by the end of training, even the smallest models always generate valid outputs (i.e. either X or Y) across the experiments we considered, suggesting that their poor performance on task disambiguation is not simply due to their poor prompting abilities.

## D   HOW IMPORTANT IS THE FORMAT OF THE UNINFORMATIVE INSTRUCTIONS?

As previously noted, our experiments on uninformative examples in the main text use an instruction format containing [CATEGORY WITHHELD]. Here we test whether the behavior of models is sensitive to variations in this phrasing. We replace [CATEGORY WITHHELD] with the linguistic nonce word `wug`, so a full instruction might read `Output 'X' if the sentence contains a wug and 'Y' otherwise`. As shown in Figure 8 we find that performance is the same for both tasks, suggesting that the particular form of uninformative instruction is not very important.

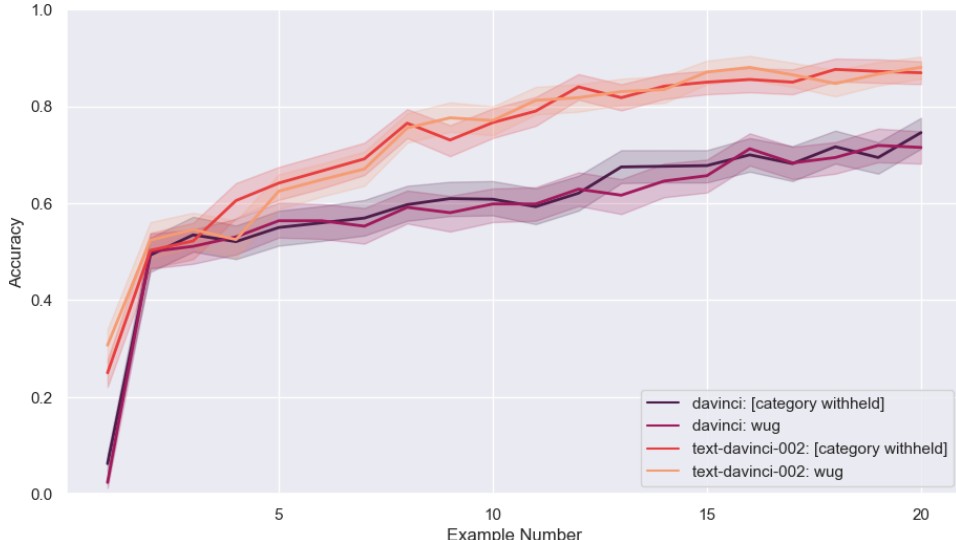

Figure 8: **The choice of format for the uninformative instructions has negligible effect.** `davinci` and `text-davinci-002` performance with different forms of uninformative instructions

| Model | Number of Parameters |
|---|---|
| OpenAI Babbage | 1.3B |
| OpenAI Curie | 6.7B |
| OpenAI Davinci | 175B |
| AI21 J1-Jumbo | 178B |
| T0++ | 11B |
| Flan-T5-XXL | 11B |

Table 2: Number of parameters for each model. Note that the OpenAI Ada and all HFD model parameter counts have not been released. It is also possible that the normal and HFD models have different numbers of parameters and were trained on different amounts and kinds of data.

## E  PARAMETER COUNTS FOR EACH MODEL

The number of parameters for each model is shown in Table 2 for Babbage,[8] Curie,[9] Davinci,[10] J1-Jumbo,[11] and T0++.[12] Note that the parameter counts of the Ada model as well as the HFD models have not been released. Furthermore, note that parameter count is not necessarily the most informative proxy for a model's capabilities, given the importance of other factors such as training data quality and the total number of tokens seen by the model (Hoffmann et al., 2022).

## F  ADDITIONAL INFORMATION ON HUMANS (PROLIFIC) EXPERIMENTS

Here we provide more information about the experiments with human participants via Prolific (Palan & Schitter, 2017).

Prior to beginning the experiment, Prolific participants were shown a message confirming consent followed by a set of boiler-plate instructions on the upcoming task which stated: 1) "Continue the pattern in the following screens to the best of your ability, using the provided instructions and examples given" 2) "Each time you make a prediction, the next screen will show the correct answer

---

[8]https://beta.openai.com/docs/model-index-for-researchers

[9]https://beta.openai.com/docs/model-index-for-researchers

[10]https://beta.openai.com/docs/model-index-for-researchers

[11]https://www.ai21.com/blog/announcing-ai21-studio-and-jurassic-1

[12]https://bigscience.huggingface.co/blog/t0

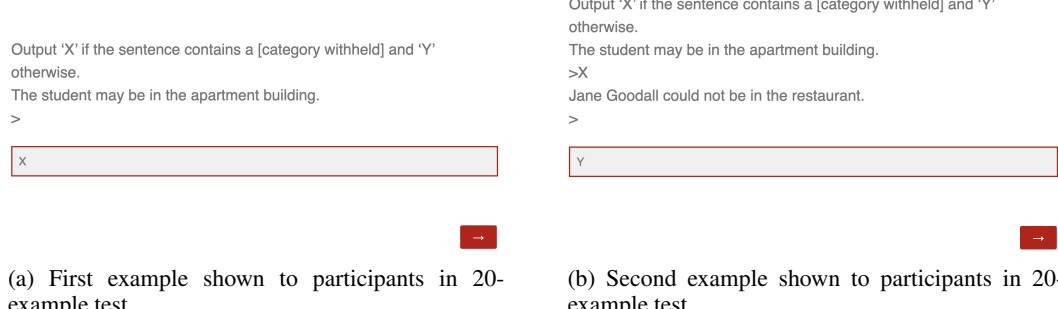

(a) First example shown to participants in 20-example test

(b) Second example shown to participants in 20-example test

Figure 9: Example questions shown to participants in Section 4.3

for that example (not necessarily your previous answer)" and 3) "The pattern may not be obvious at first but may become more clear over time. Note: some information in the instructions may be [intentionally withheld] (denoted by brackets)" This prelude was deemed necessary after a series of pilot runs in which participants quit out of the survey due to their confusions and/or frustrations with the lack of guidance. To further streamline the process, participants were also given a set of useful keyboard shortcuts with which to progress through the survey (although participants were not allowed to edit previous answers).

For the tests in Section 4.3, participants were shown examples one-by-one with the correct answer appearing on the screen following their input. For the tests in Section 4.2 tests, participants were shown both labeled examples and the one unlabeled query example at the same time.

For the tests in Section 4.3, each participant was only given one set of 20 examples. For the tests in Section 4.2, each participant was only given a single prompt. Prompts given to participants were randomized and roughly evenly distributed across all tests. Participant IDs were filtered to ensure that the same participant did not participate multiple times across studies. All surveys were created using Qualtrics' survey builder.

### F.1 NUMBER OF PARTICIPANTS FOR HUMAN EXPERIMENTS

**Task disambiguation using natural language instruction** For the experiment referenced in Section 4.2 we recruited 51 and 33 participants for the unambiguous instruction and informative instruction tests respectively.
**Task disambiguation using multiple examples** For the experiment detailed in Section 4.3, we recruited 96 total participants: 16 per salient feature.

## G AMBIBENCH BENCHMARK: FULL DETAILS

AmbiBench consists of ambiguous sentences with two explicitly varying (salient) features in each sentence. Salient features are grouped into pairs: 'subject' and 'location, 'religious' and 'pronoun', and 'proper noun' and 'negation.' Each salient feature has two values (e.g. *animal subject* or *human subject*) with each choice having a 0.5 probability of being selected.

### G.1 SALIENT FEATURE PAIRINGS

The pairings of the salient features are detailed below. The pairings remained constant throughout all experiments.

#### G.1.1 SUBJECT-LOCATION SENTENCES

Sentences follow the template: "The {*human/animal subject*} is in the {*indoor/outdoor location*}." The subject is randomly assigned to either a human or an animal. If chosen to be a human, the

subject is randomly chosen from a list of professions: *[student, reporter, hiker, researcher, firefighter, fugitive, critic, photographer, director, surveyor]*. If chosen to be an animal, the subject is randomly chosen from a list of common animals: *[boar, worm, hawk, hound, butterfly, snake, duck, bear, mountain lion, horse]*. The location is randomly assigned to either an outdoor location or an indoor location. If chosen to be an outdoor location, the location is chosen from a list of common outdoor locations: *[river, pond, woodlands, cave, canyon, prairie, jungle, marsh, lagoon, meadow]* whereas if chosen to be an indoor location, the location is chosen from a list of common indoor locations: *[laboratory, theatre, museum, courtroom, apartment building, restaurant, house, film studio, hotel lobby, grocery store]*.

### G.1.2 RELIGIOUS-PRONOUN SENTENCES

Sentences follow the template: "{*She/He*} is in the {indoor location} with the {*religious/secular leader*}." The pronoun is randomly assigned to either a *He* or *She*. The leader is randomly assigned to either a religious leader or a secular leader. If chosen to be an religious leader, the leader is chosen from a list of common religious leaders: *[pope, reverend, bishop, Dalai Lama, rabbi, cardinal, pastor, deacon, imam, ayatollah]* whereas if chosen to be a secular leader, the leader is chosen from a list of common secular leader: *[president, CEO, principal, sheriff, judge, ambassador, officer, prime minister, colonel, professor]*. The indoor location in the sentence is randomly chosen from the aforementioned list of indoor locations.

### G.1.3 PROPER NOUN-NEGATION SENTENCES

Sentences follow the template: "{*Proper/Common noun*} {*negation/affirmation*} in the {indoor location}." The noun is randomly assigned to either a proper noun or a common noun. If chosen to be a proper noun, the noun is randomly chosen from a list of famous individuals across a variety of disciplines: [*Lebron James, Bernie Sanders, Christopher Nolan, Paul Atreides, Noam Chomsky, Serena Williams, Margot Robbie, Alexandria Ocasio-Cortez, Hermione Granger, Jane Goodall*]. If chosen to be a common noun, the noun is randomly chosen from the prior list of human subjects: [*student, reporter, hiker, researcher, firefighter, fugitive, critic, photographer, director, surveyor*] and appended to 'The' to maintain grammaticality of the sentence. The verb is randomly assigned to either a negation or an affirmation. If chosen to be a negation, the verb is chosen from a list of common negation verb phrases: *[is not, was not, has not been, may not be, could not be]*. If chosen to be an affirmation, the verb is chosen from the list of affirmations directly contrasting the list of negations: *[is, was, has been, may be, could be]*.

## G.2 INSTRUCTIONS

Instructions given were either informative instructions, explicitly stating the salient feature which determined the label for each sentence, or uninformative instructions which did not divulge the salient feature. Informative Instructions were of the template: "Output 'X' if the sentence {*salient feature instruction*} and 'Y' otherwise", where *salient feature instruction* could be filled in with one of {contains a reference to an indoor/outdoor location, contains a reference to a human/an animal, contains a reference to a religious leader, does not contain a reference to a religious leader, contains a reference to a religious leader, contains a male pronoun, contains a female pronoun, contains a proper noun, does not contain a proper noun, contains a negation, does not contain a negation}

Uninformative instructions were always given as "Output 'X' if the sentence contains a [category withheld] and 'Y' otherwise."

## G.3 TASK DISAMBIGUATION USING NATURAL LANGUAGE INSTRUCTIONS

When constructing tests for Section 4.2, we ensured task ambiguity by grouping one possibility for a given salient feature with one possibility for its paired salient feature.

For example, for Subject-Location sentences, humans and indoor locations were grouped and animals and outdoor locations were grouped.

In this test, humans and models were given one example with the first of these groupings and a second example with the second of the groupings. The final, 'disambiguating,' example broke the

groupings, therefore displaying which of the features was responsible for the example's label (the salient feature).

Here is a demonstration of a single prompt for an example with the Subject-Location sentence type:

Q: The hawk is in the canyon.
A: X
Q: The director is in the museum.
A: Y
Q: The hiker is in the meadow.
A: X

For the first two examples, it is unclear whether the subject or the location is controlling the label but upon seeing the third example, it becomes clear that the location and not the subject is controlling the label (as the sentence is labeled X whenever an outdoor location is referenced).

Tests contained either an uninformative instruction or an informative instruction followed by three examples. Humans and models were shown all three examples at once. Humans were shown the first two examples with labels and the last example without the label. Models were shown all three examples with labels within the same API query (as the log probabilities for the final label can be acquired without requiring a completion).

The order of groupings, order of labels, labels' correspondence with a salient feature, and the salient feature were randomized across all tests with equal probabilities being given to each possibility.

Some example prompts from the test set:

**Prompt 1**. Salient feature is 'subject' (Q/A format with informative instructions):
```
Output 'X' if the sentence contains a reference to a human and 'Y'
otherwise.
The horse is in the woodlands.
A: Y
Q: The student is in the laboratory.
A: X
Q: The mountain lion is in the film studio.
A: Y
```

**Prompt 2**. Salient feature is 'pronoun' (arrow format with uninformative instructions):
```
Output 'X' if the sentence contains a [category withheld] and 'Y'
otherwise.
He is in the film studio with the imam.
>Y
She is in the restaurant with the judge.
>X
She is in the apartment building with the bishop.
>X
```

**Prompt 3**. Salient feature is 'proper noun' (Q/A format with uninformative instructions):
```
Output 'X' if the sentence contains a [category withheld] and 'Y'
otherwise.
Q: The student was not in the hotel lobby.
A: X
Q: Lebron James is in the theatre.
A: Y
Q: Christopher Nolan could not be in the house.
A: Y
```

## G.4   TASK DISAMBIGUATION USING MULTIPLE EXAMPLES

When constructing tests for Section 4.3, humans and models were shown a set of 20 questions with the same salient feature. All sentences within the set contained the same sentence pairing Subject-Location, Religious-Pronoun, Proper Noun-Negation.

For each example, the possibilities for each of the two features were randomized and given equal probability (as described in the process of sentence construction). The groupings mentioned for the tests in Section 4.2 were not used.

The salient feature was randomly selected (with each possibility being given an equal probability).

Tests always contained an uninformative instruction prior to the set of examples.

Humans were shown questions one at a time, progressively building up from 1 to 20 questions. After each question, the correct answer was displayed on the screen (in place of their solution if incorrect). Models were shown all 20 questions within the same API query and the log probabilities for each of the twenty labels were tracked.

As with the tests in Section 4.2, the labels' correspondence with a salient feature and the order of labels were randomized across all tests with equal probabilities being given to each possibility.

Some example prompts from the test set:

**Prompt 1**. Salient feature is 'location' (arrow format):
```
Output 'X' if the sentence contains a [category withheld] and 'Y'
otherwise.
The photographer is in the restaurant.
>Y
The mountain lion is in the river.
>X
The hiker is in the cave.
>X
The butterfly is in the grocery store.
>Y
The surveyor is in the river.
>X
The boar is in the prairie.
>X [continues until 20 examples]
```

**Prompt 2**. Salient feature is 'religious' (Q/A format):
```
Output 'X' if the sentence contains a [category withheld] and 'Y'
otherwise.
Q: He is in the film studio with the ayatollah.
A: X
Q: She is in the house with the CEO.
A: Y
Q: He is in the apartment building with the ambassador.
A: Y
Q: She is in the museum with the rabbi.
A: X
Q: She is in the museum with the Dalai Lama.
A: X
Q: He is in the laboratory with the rabbi.
A: X [continues until 20 examples]
```

**Prompt 3**. Salient feature is 'negation' (arrow format):
```
Output 'X' if the sentence contains a [category withheld] and 'Y'
otherwise.
The student was not in the restaurant.
>X
Christopher Nolan was in the apartment building.
>Y
The photographer has not been in the theatre.
>X
Alexandria Ocasio-Cortez could be in the film studio.
>Y
Christopher Nolan was not in the museum.
```

```
>X
The photographer was in the film studio.
>Y [continues until 20 examples]
```

## G.5 Finetuning a Model to Generalize in the Face of Task Ambiguity

When constructing the finetuning dataset for Section 4.4, two out of the three possible sentence pairings were used. We then tested on the held-out sentence pairing. For example, if the finetuning dataset contained the salient features: 'subject,' location,' 'religious,' and 'pronoun,' the held-out features would be 'proper noun' and 'negation.'

For the control experiments, the dataset consisted of examples for which only one feature varied. For example, if the sentence was of the Subject-Location sentence type and the salient feature was location, a given example may contain examples with both outdoor and indoor locations but only humans.

For the ambiguous experiments, the dataset consisted of examples with ambiguity: both features varied between the two possibilities for each feature.

The number of examples in each prompt varied: building from 4 examples to 20 examples for each salient feature. But the examples across these different prompts did not remain constant (rather just the salient feature did), unlike in Section 4.3 where a prompt with a length of n examples simply added one example to the preceding prompt of length n-1.

As with the tests in Section 4.2 and Section 4.3, the labels' correspondence with a salient feature and the order of labels were randomized across all tests with equal probabilities being given to each possibility.

## H Additional Information about Finetuning Experiments

**Hyperparameters**  We use OpenAI's finetuning API to finetune their `davinci` model.[13] When conducting the finetuning, we used a batch size of 1, a learning rate multiplier of 0.1, and a prompt loss weight of 0.1.

**Control experiment settings**  Our control experiment for the finetuning only varied one of the two features in each set of 20 examples (e.g. *boar* could change to *worm* but not *firefighter*). The choice of which feature would be constant (e.g. *human/animal* vs *indoor/outdoor*), as well as the value of that feature (e.g. *human* vs *animal*) were decided randomly for each set of 20 examples. By following this procedure, we did not introduce task ambiguity into the finetuning data, but still ensured that the control model saw the same range of sentence constructions as the model trained on ambiguous examples.

## I Additional Graphs

### I.1 Task Disambiguation Using Natural Language Instructions

#### I.1.1 Informative Instructions

Figure 10 details the performance across salient features for each model individually across both format types in instances of informative instructions. Figure 11 averages the performance of the models across salient features, demonstrating the difference in performance by format type for each salient feature.

#### I.1.2 Uninformative Instructions

Figure 12 and Figure 13 mirror Figure 10 and Figure 11 respectively but instead demonstrate the performance in prompts with uninformative instructions.

---

[13]https://beta.openai.com/docs/guides/fine-tuning

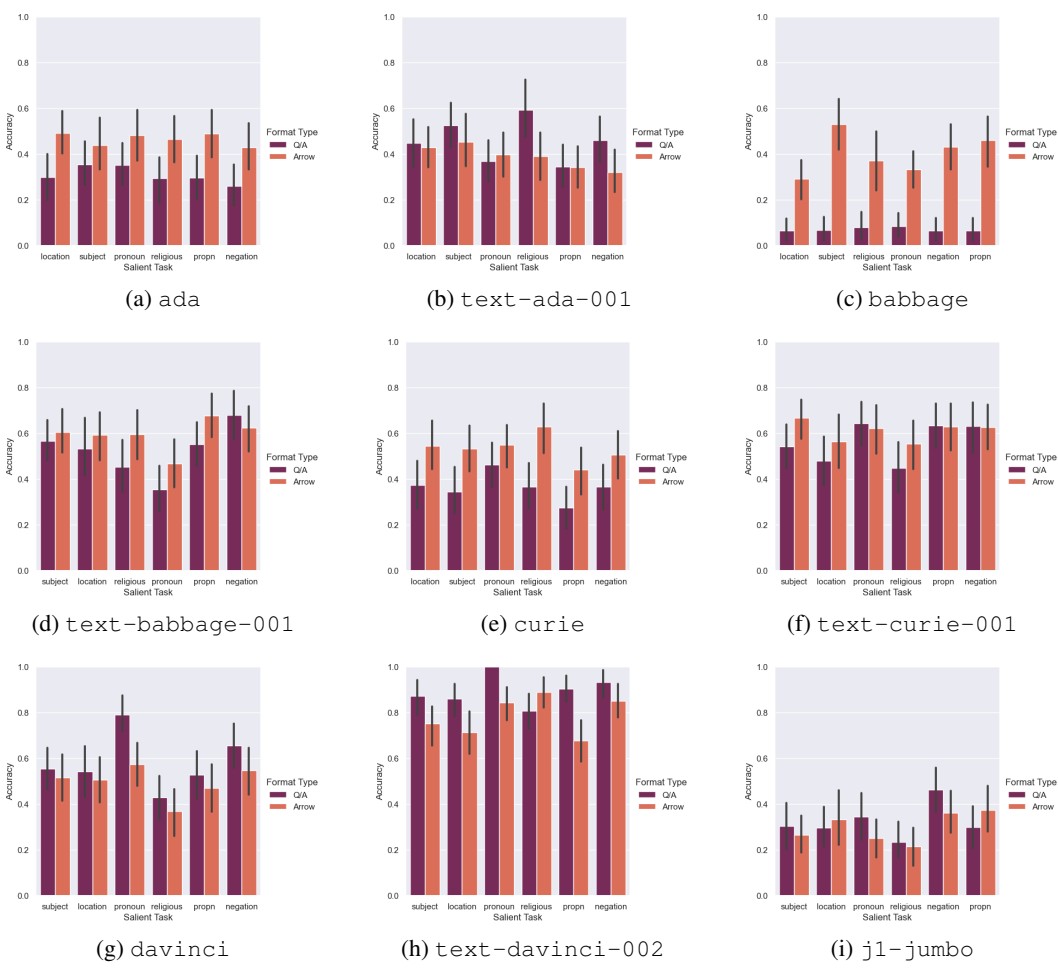

Figure 10: Performance with informative instructions across salient features on Task Disambiguation Using Natural Language Instructions (*see Section 4.2*)

## I.2 TASK DISAMBIGUATION USING MULTIPLE EXAMPLES

Figure 14, Figure 15, Figure 16, Figure 17, and Figure 18 individually detail the performance of each model across each salient feature from of 1-20 examples.

Figure 19b demonstrates the difference in performance across the two format types, averaged for all models. Because of t0pp's poor performance on prompts with the arrow format, we do not include t0pp in the collective performance graph but instead display its performance across formats individually from the other models.

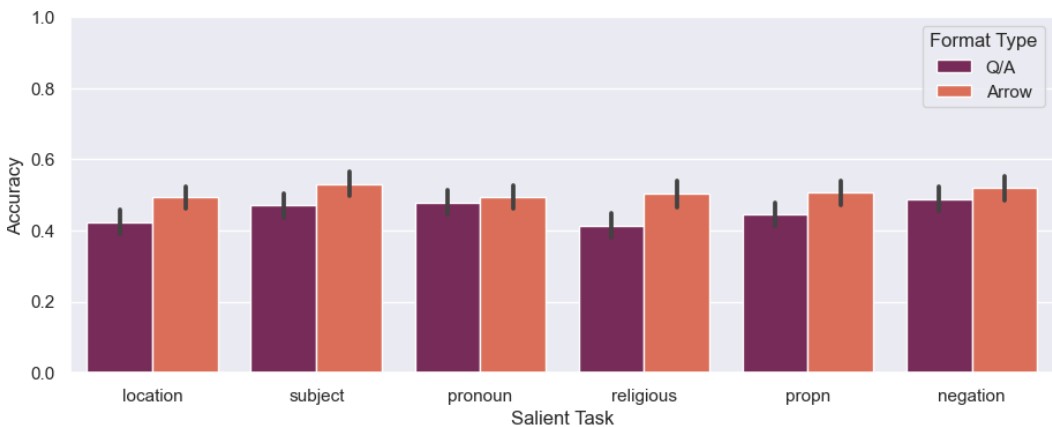

Figure 11: Performance with informative instructions for each format averaged over all models on Task Disambiguation Using Natural Language Instructions (*see Section 4.2*)

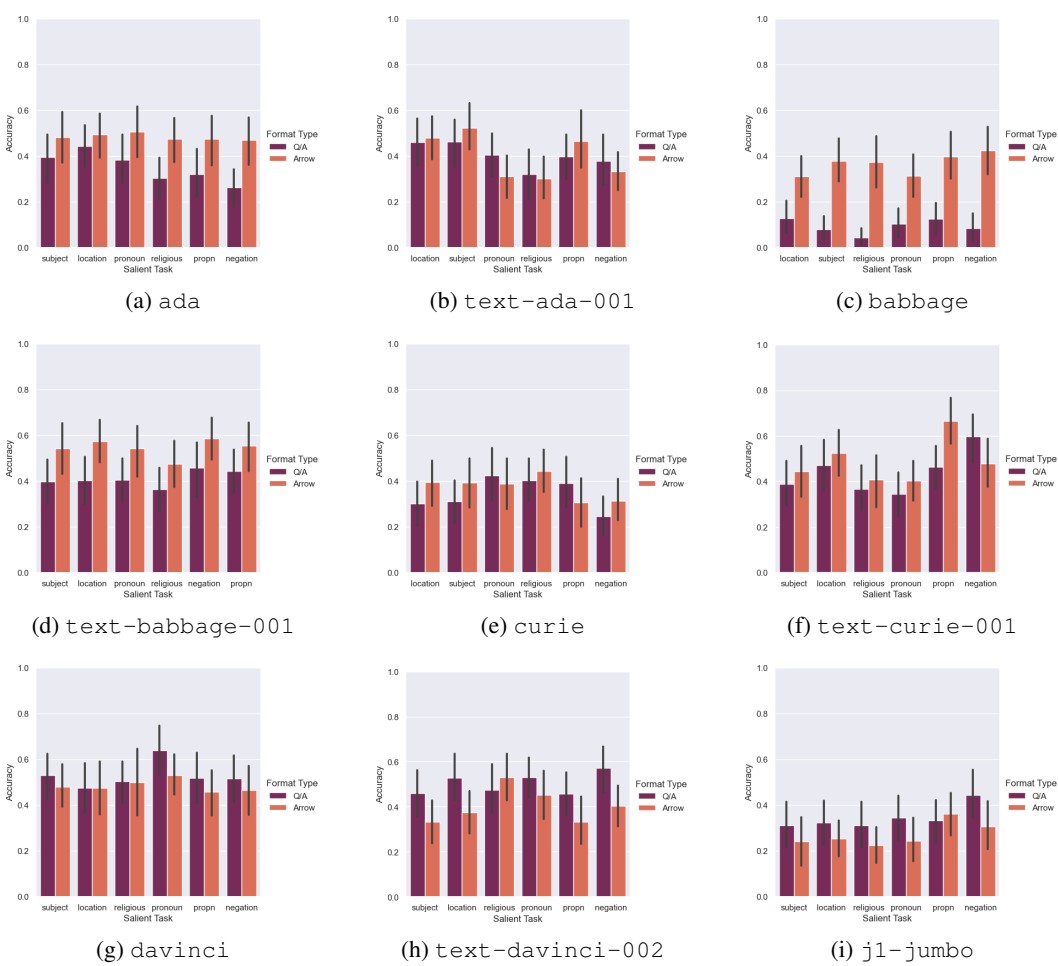

Figure 12: Performance with uninformative instructions across salient features on Task Disambiguation Using Natural Language Instructions (*see Section 4.2*)

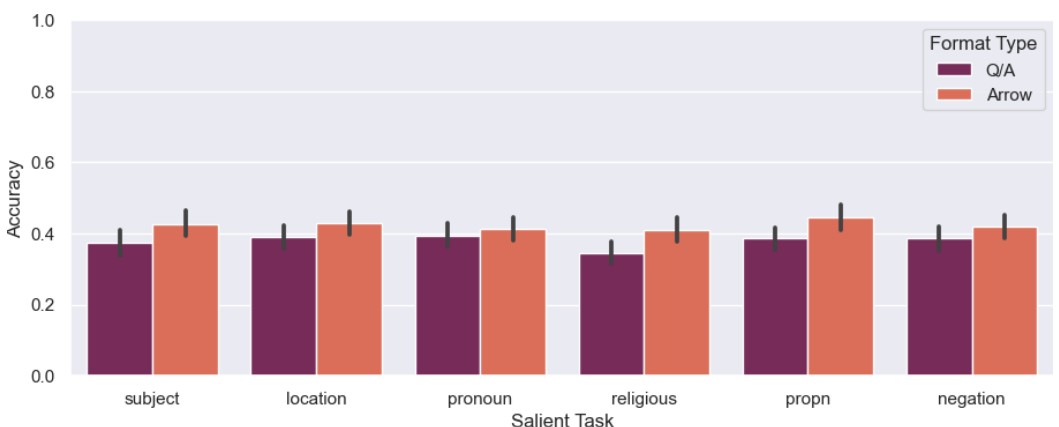

Figure 13: Performance with uninformative instructions for each format averaged over all models on Task Disambiguation Using Natural Language Instructions (*see Section 4.2*)

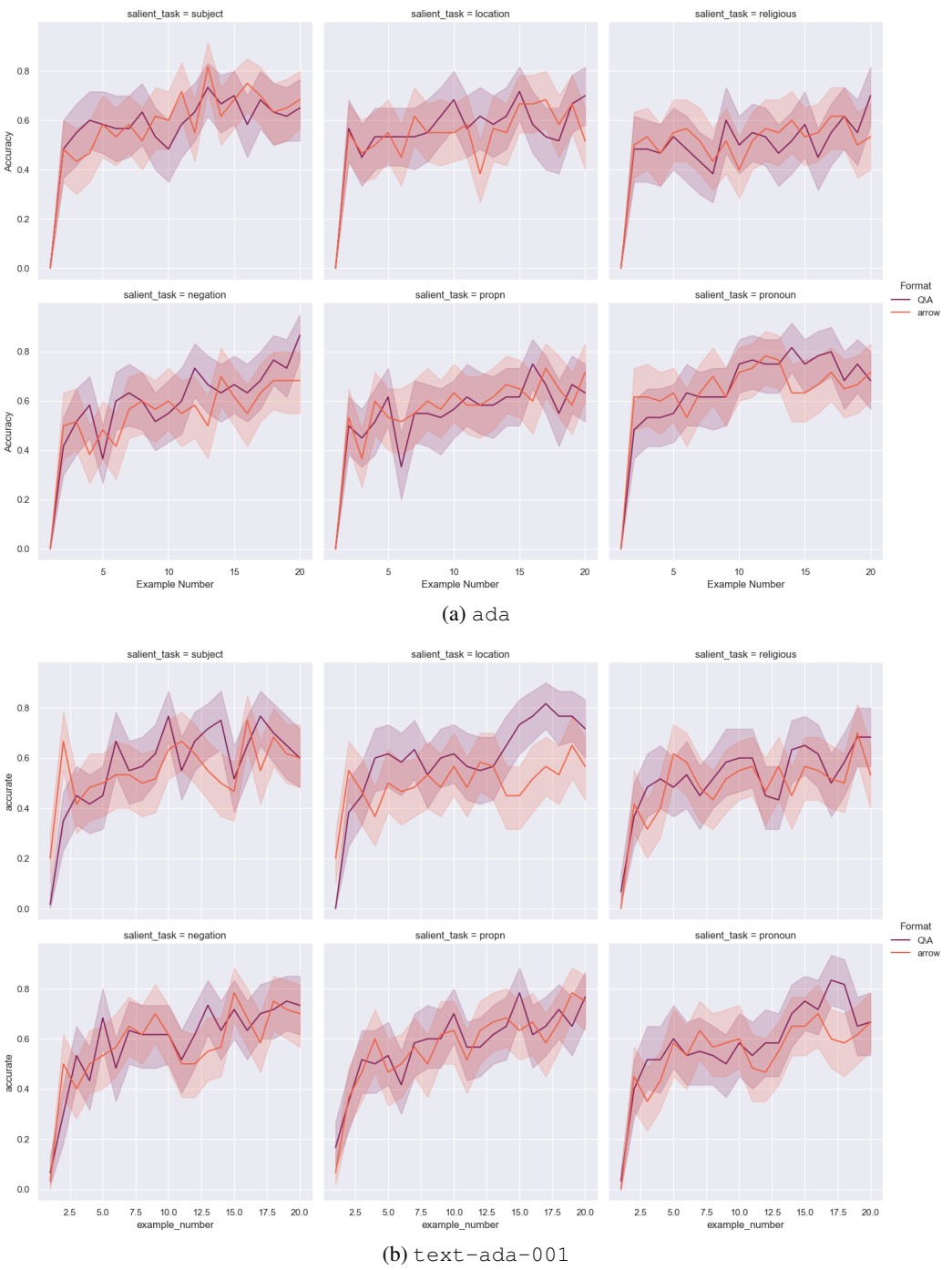

(a) `ada`

(b) `text-ada-001`

Figure 14: Performance across salient features on Task Disambiguation Using Multiple Examples (*see Section 4.3*)

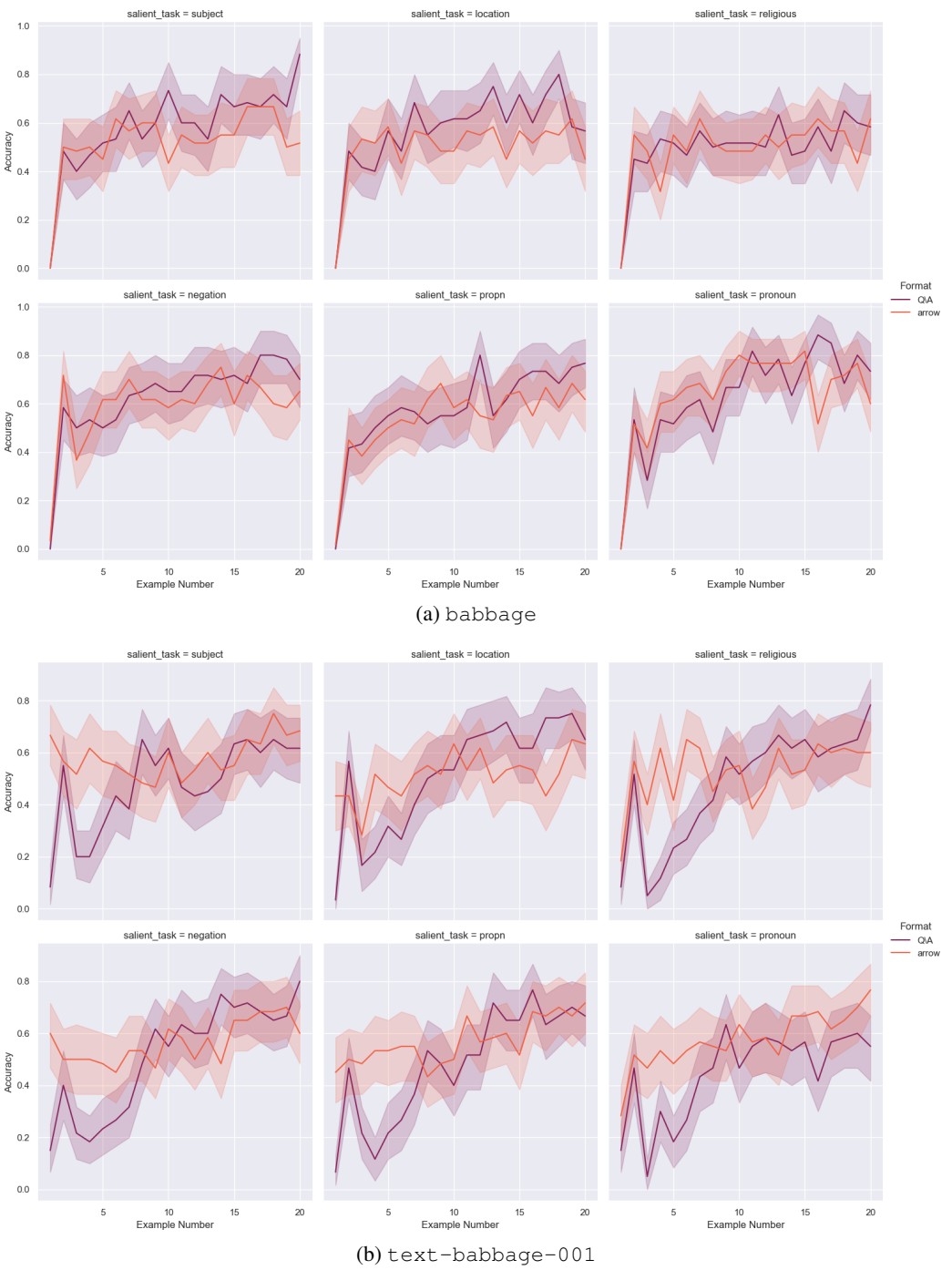

(a) `babbage`

(b) `text-babbage-001`

Figure 15: Performance across salient features on Task Disambiguation Using Multiple Examples (*see Section 4.3*)

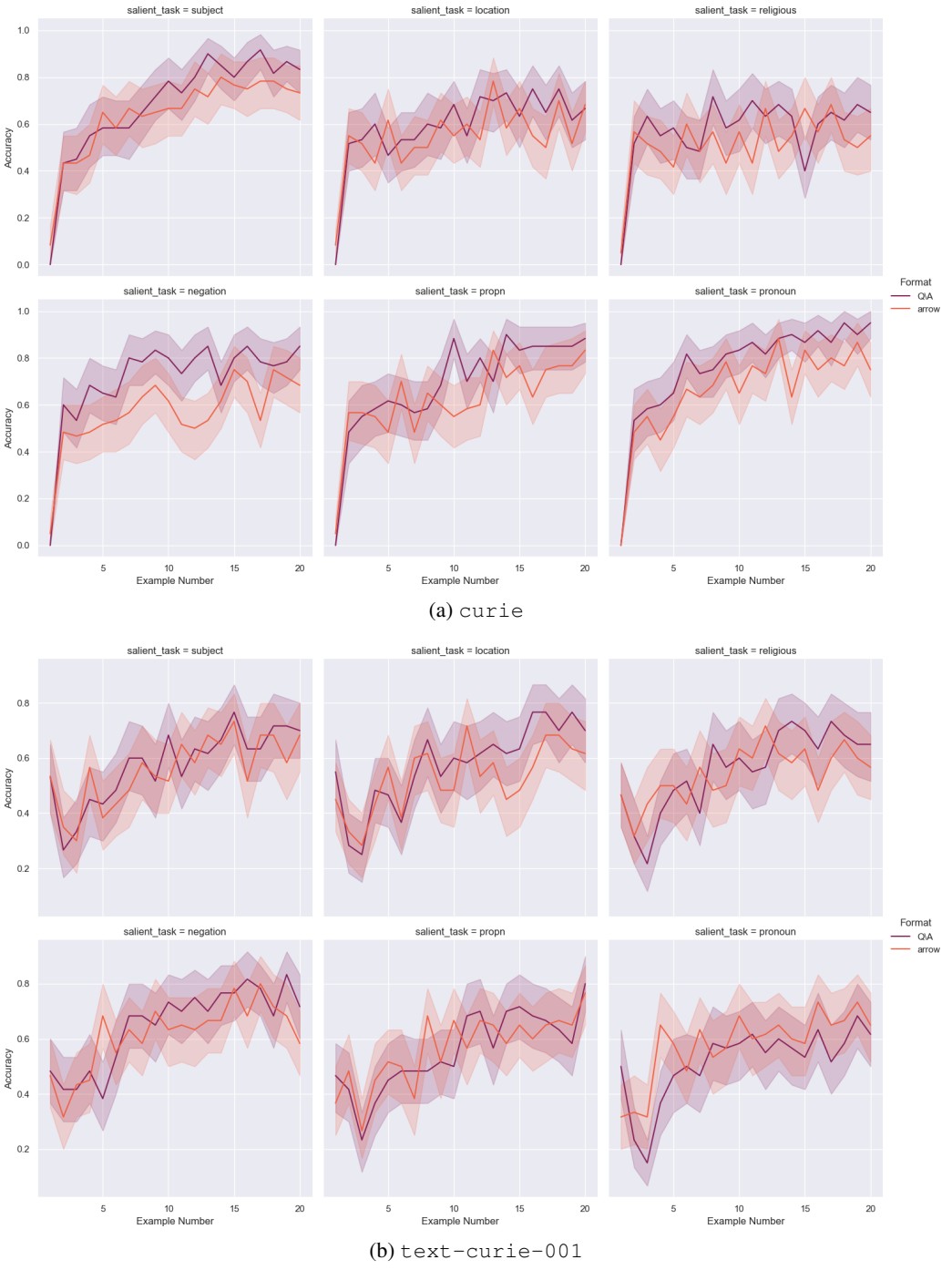

(a) `curie`

(b) `text-curie-001`

Figure 16: Performance across salient features on Task Disambiguation Using Multiple Examples
(*see Section 4.3*)

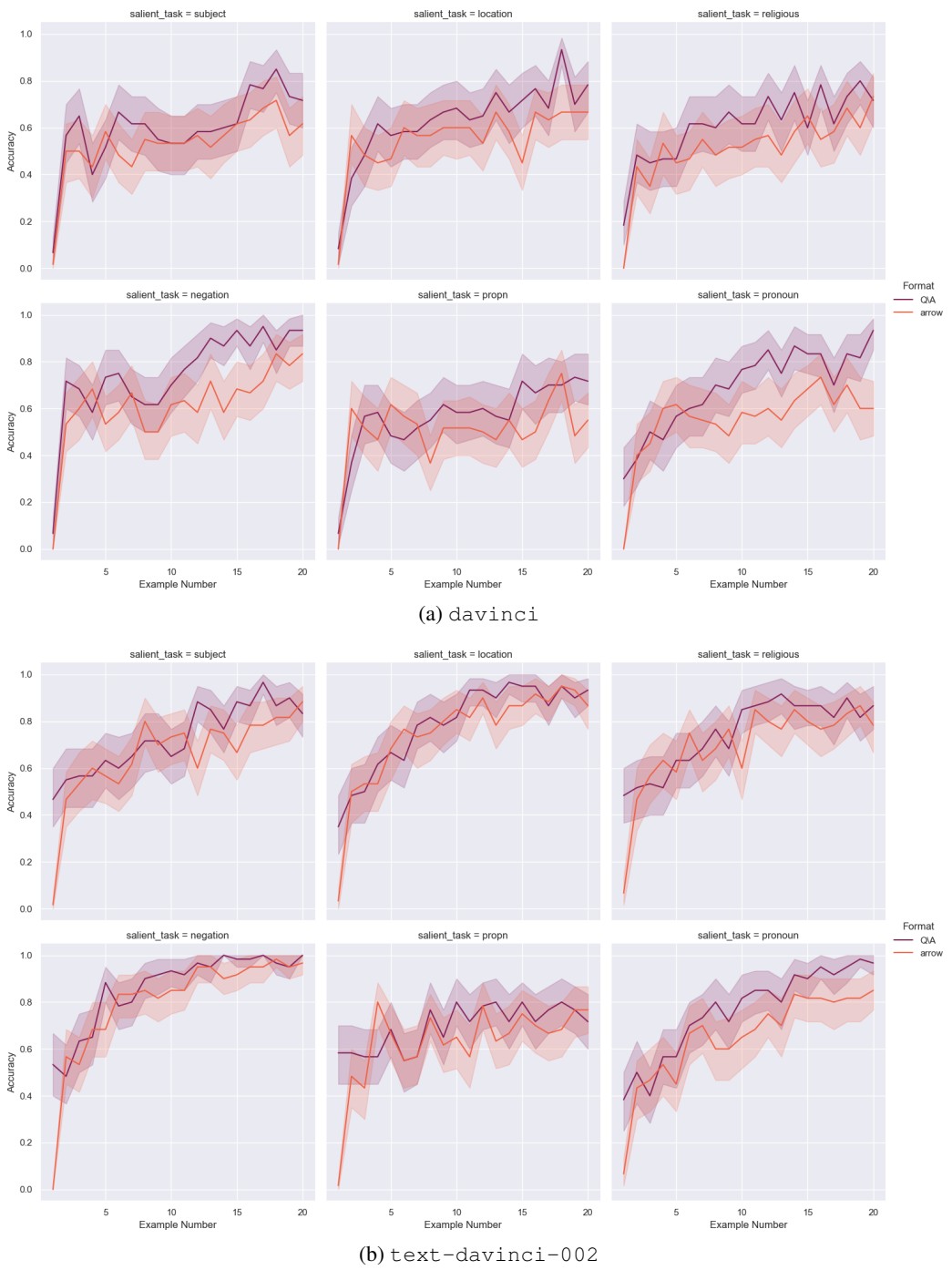

(a) davinci

(b) text-davinci-002

Figure 17: Performance across salient features on Task Disambiguation Using Multiple Examples (*see Section 4.3*)

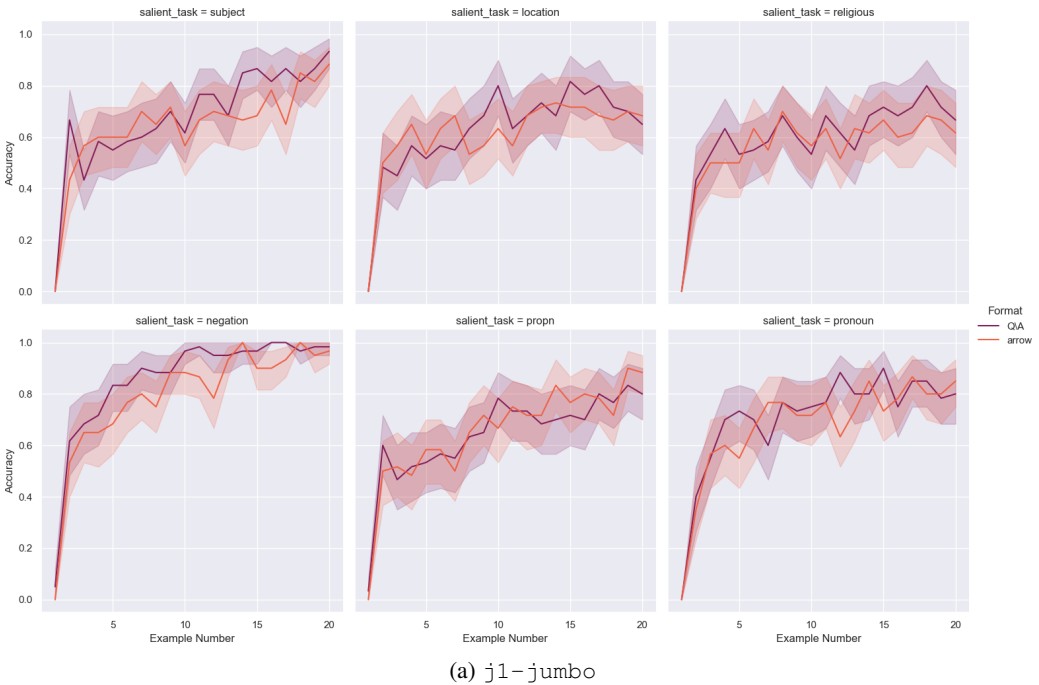

(a) `j1-jumbo`

Figure 18: Performance across salient features on Task Disambiguation Using Multiple Examples (*see Section 4.3*)

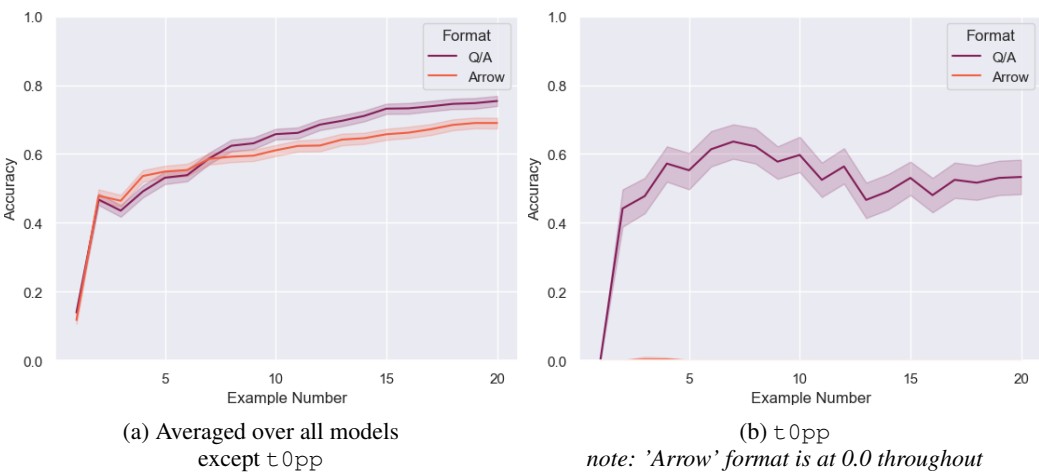

(a) Averaged over all models except `t0pp`

(b) `t0pp`
*note: 'Arrow' format is at 0.0 throughout*

Figure 19: Performance of each format on Task Disambiguation Using Multiple Examples (*see Section 4.3*)

