# OpenReview forum: "Task Ambiguity in Humans and Language Models"
_ICLR.cc/2023/Conference — ICLR 2023 poster_

### Official Review · Reviewer_o1Uh · 2022-10-19

**Confidence:** 4
**Correctness:** 3
**Technical Novelty And Significance:** 2
**Empirical Novelty And Significance:** 3
**Recommendation:** 8

**Clarity, Quality, Novelty And Reproducibility:**

Let me list my concerns and questions here, please answer and help me better understand the paper :)

1. In Section 2.2, the author mentioned a few prior works discussing LLMs' unfavourable behaviours such as "recency bias" as proposed in [1]. Indeed, for most in-context learning or few-shot prompting work, such biases can have non-negligible effects on model performance (e.g., models tend to copy labels from the most recent examples as its prediction, due to such effects, model performance on a data point can sometimes vary from random to near perfect). The authors state that \
`In this work, we attempt to control for such factors by constructing a benchmark where multiple tasks are consistent with a single instruction or example, requiring the model to leverage multiple signals to disambiguate the intended task without relying on learned priors.` \
Please help me understand this, I fail to understand how AmbiBench avoids or mitigates such effects? To my understanding, as long as few-shot prompting are provided as a sequence, such effects can exist.
2. What are some reasons in Figure 2 and Figure 3 human annotator scores have a much bigger variance?
3. I like how the authors include many LLM variants for comparison. However, I'm a bit concerned on, e.g., in Section 4.2.1, what's a good way to decouple models' 1) inability of disambiguate the task description from 2) inability of generate meaningful output via prompting (maybe a model can do disambiguation, but it's not big enough to perform well on prompting?) Given some recent discussions such as [2], is few-shot prompting the best way to investigate certain abilities of smaller models? Compared to, e.g., using a linear/non-linear probe to see if the LM representations contains certain properties we want. If the LM has trouble to even generate meaningful outputs (as reported by the authors, sometimes models generates tokens other than `X` and `Y`), it's a bit hard to tell how capable the model is on task disambiguation.
4. I'd like to hear more from the authors about the fine-tuning experiments. I assume that fine-tuning could potentially make the disambiguation task much easier, even with only a small set of training data (68 per task). I fail to gain too much information from Section 4.4 more than fine-tuning on in-context data can outperform a general inference-only model.
5. As the authors mention, AmbiBench is a *minimal complexity* testbed. I understand and agree that this is to make the benchmark more controllable. However, for the same reason, I am less convinced about the generality of findings reported here. What are some other types of task ambiguities in NLP tasks, maybe more subtle than simply occluding a category name? Maybe brought by natural language instructions rather than templatic language? What about in tasks other than sentence classification? It would be nice to have such discussion, which can give readers a better idea how the proposed method/benchmark can shed light on understanding task disambiguation in general.
6. (less important:) Figure 1 is a bit misleading in some sense, I was honestly expecting to see how a LLM can explicitly guess the task description --- this might not be a crazy idea, given a sequence of examples (as many as 20 here), is there a way to model/measure such thing?
7. (less important:) The two prompt templates introduced in Section 3.3 are quite arbitrary, especially the `Q/A` format barely makes sense.


### References:
* [1] [Calibrate Before Use: Improving Few-Shot Performance of Language Models. Zhao et al., 2021.](https://arxiv.org/pdf/2102.09690.pdf)
* [2] [Emergent Abilities of Large Language Models. Wei et al., 2022.](https://arxiv.org/pdf/2206.07682.pdf)

**Strength And Weaknesses:**

### Strength

* Interesting and useful topic: I really like seeing work on this topic. Task ambiguity has never been a more important topic thanks to the flourishing of LLM research. In prompt-based learning where users/practitioners can use an inference-only LLM and prompt it to perform some never-before-seen tasks. At the same time of benefiting from greater accessibility, users have to describe their tasks (in language, as part of the prompt) in a way that maximizes LLM performance. I see this work quite related to the line of work of prompt engineering, but tackling a very specific aspect of prompting LLMs. Although using a toy-ish benchmark, I can see potential empirical value from such work.
* Human performance: I really appreciate the authors who include human scores for the tasks, which greatly helped me get a clearer picture. Although the second paragraph of Section 4.3.1 (why text-davinci-002 outperforms humans in Figure 3) seems a bit hand wavy to me.
* Clarity: I enjoy reading this paper, most of the content are easy to follow.



### Weaknesses

Please see my comments below.

**Summary Of The Paper:**

In this paper, the authors investigate the problem of task ambiguity in prompting pre-trained LLMs. In which setting, the task descriptions sometimes contain insufficient information for humans/models to perform the task, they have to consequently look at a few examples to have a better guess on the task. The authors propose a new benchmark designed for studying the said problem, namely AmbiBench, which includes six manually designed sentence classification tasks. Using AmbiBench, they tested a set of pre-trained LLMs (covering a wide spectrum of training data sizes, training procedures, and model architectures) as well as a group of human annotators. Results suggest that only the largest LLM variant trained with additional learning from human feedback (text-davinci-002) can approach human performance on such tasks. Further, the authors show that the davinci model can quickly improve if fine-tuned on a small amount of in-context examples, even outperforming the text-davinci-002 model.

**Summary Of The Review:**

In general, I think the current version of the paper is around (maybe slightly above) the borderline. I look forward to reading the authors responses to my questions so I can better understand the work and thus have a more precise evaluation.

---

> ### Author Response · Authors · 2022-11-14
> **Response [1/2]**
>
> We thank the reviewer for their review! We are glad they appreciate the importance of studying task ambiguity, the empirical value of the work, and the inclusion of human studies.
>
> **Q1) How does AmbiBench control for different effects observed in language models?**
>
> That phrase "we attempt to control for such factors" refers to the factors mentioned in the same paragraph (the second paragraph of 2.2). We have rewritten the second paragraph of Section 2.2 to make this more clear.
>
> To provide some additional detail: [3] finds that language models can ignore the content of their few-shot examples, relying more on the format of the prompt and the tasks they've seen in the training data. Our task setup ensures that language models that perform well *must* do so by using the information from the few-shot examples, since 1) Our task is new, and 2) there are multiple possible tasks they might be asked to perform.
>
> Other effects such recency bias (e.g. as mentioned in the first paragraph of 2.2) are certainly still important and affect our experiments. We attempt to mitigate them a few different ways, such as by generating new examples each time and shuffling their order.
>
>
> [3] Sewon Min, Xinxi Lyu, Ari Holtzman, Mikel Artetxe, Mike Lewis, Hannaneh Hajishirzi, and Luke Zettlemoyer. Rethinking the role of demonstrations: What makes in-context learning work?
> [4] Albert Webson and Ellie Pavlick. Do prompt-based models really understand the meaning of their prompts?
>
> **Q2) Why are the human error bars larger than the models?**
>
> Overall, we recruited 180 participants (Appendix F1), whereas we queried each model an order of magnitude more times. This was primarily due to financial reasons—it is far cheaper to query a model than to recruit a human participant (we paid participants at $12-13 / hr).
>
> **Q3) Disentangling models prompting abilities from their ability to disambiguate tasks**
>
> This is a good point. In some sense, the model's ability to disambiguate tasks is an aspect of its ability to be prompted well, since a key component of prompting is how quickly the model can identify the intended task. But perhaps one could look at the correlation between some overall prompting score and the task ambiguity score to find models which "punch above their weight" as far as task ambiguity goes. We have added some discussion to this end in Appendix C and Figure 7, where we find that at the end of training all models, even the smallest models (e.g. ada, babbage) almost exclusively generate valid outputs (either X or Y). This indicates they are able to be prompted well enough to produce reasonable outputs for the task.
>
> **Q4) More explanation of why finetuning experiments are interesting**
>
> Our finetuning experiments aren't the typical kind of finetuning where you finetune a model on a task then evaluate the model on other examples of that same task. Instead we explore fine-tuning as a way to implement *meta-learning* for *new* in-context tasks. Furthermore, we show that the naive approach of training on normal in-context examples actually yields no benefit for resolving task ambiguity. However, if one trains on *ambiguous* in-context tasks, this improves a model's ability to learn well via few-shot learning even on *unseen* ambiguous tasks, so much so that it outperforms the best models trained with reinforcement learning from human feedback. That is, our meta-training approach creates a model that is more capable *in general* of handling task ambiguity in unseen tasks.

---

> > ### Comment · Reviewer_o1Uh · 2022-11-18
> > **Thank you**
> >
> > I have read other reviewers' comments as well as the authors' response. Overall I am happy with the authors' response to my questions and concerns, they are clear. I also appreciate the authors' effort adding the new Appendix A-D, they really helped me better understanding the work.
> >
> > I'd like increase my score a bit, but since we don't have the option of 7, I'll give an 8.

---

> ### Author Response · Authors · 2022-11-14
> **Response [2/2]**
>
> **Q5) Other kinds of task ambiguity**
>
> We also believe it would be interesting to explore other kinds of task ambiguity. To that end, we have added a section to the paper (Appendix B) about how one might expand our studies to a second, more realistic kind of task ambiguity. Specifically, we consider an ambiguously-specified task of parsing an utterance into a command line invocation. Here is an example prompt:
>
> ```
> Write the AWS CLI command to create an AWS Bucket.
>
> Input: Create a bucket for Sato Tamotsu
> Output: aws s3 mb s3://bucket-for-sato --region ap-northeast-1
>
> Input: Create a bucket for Yuki Hashimoto
> Output: aws s3 mb s3://bucket-for-yuki --region ap-northeast-1
>
> Input: Create a bucket for Margaret Richards
> Output:
> ```
>
> Unlike the two-feature ambiguity we consider in AmbiBench, here the ambiguity comes from whether the system should continue the pattern, creating the AWS bucket in the ap-northeast-1 region (Japan), or make the assumption based on the name Margaret Richards that the bucket should be placed in another region (e.g. in Europe or the USA).
>
> We ran some initial experiments in this setting, demonstrating that indeed language models behave differently in terms of which hypothesis they favor. We also note how similar meta-learning via fine-tuning experiments as we explore could teach models the desirable generalization rule, such as "when in doubt, don't make assumptions based on people's names."
>
> We thank the reviewer for this suggestion, as we believe it strengthens the paper by making clear how the cleaner synthetic experiments we explore could be extended into more real-world examples of ambiguity.
>
> **Q6) Can the language model guess the task description?**
>
> This is a very interesting question! We previously conducted some preliminary experiments here, and included them in the newest revision of the paper (Appendix A). Across the six features, only for one of the features is text-davinci-002 able to reliably guess the task description, and for one more it is able to achieve 50% correctness. This suggests that even when models can perform a task well, there is substantial headroom in getting models to reveal that knowledge in language.
>
> **Q7) Choice of prompt templates**
> We agree that the Q/A format seems less intuitive than the arrow format, however it is noteworthy that it achieves reliably better performance across models (See Figure 19a). As far as we can tell, it is an open question in the community how to choose good templates for few-shot learning, and we are hopeful that future work will produce better standards that alleviate much of this confusion.

---

### Official Review · Reviewer_zof8 · 2022-10-27

**Confidence:** 5
**Correctness:** 2
**Technical Novelty And Significance:** 1
**Empirical Novelty And Significance:** 2
**Recommendation:** 3

**Clarity, Quality, Novelty And Reproducibility:**

This paper is clear, although the writing can be more fluent and concise.

The novelty is limited. The fine-tuning methods are standard practice in this domain.

The reproducibility is unclear, as the authors do not mention whether the benchmark will be released.

**Strength And Weaknesses:**

Strength:

Natural language prompts have been extensively studied, and the ambiguity of certain tasks, or the difficulty of designing prompts that can clearly describe the task, is an important challenge that may limit the utility of these methods in the real world.

Weakness:

This paper only studies the ambiguity of a few toy tasks, which only require simple lexical understanding and the texts are synthetic and very short. It would be more interesting to see what the ambiguity is in real-world NLP tasks, and how they can be handled.

Besides, the empirical observation and proposed fine-tuning method are pretty straightforward and not surprising. The "uninformative instructions" are very unnatural as they just replace the salient feature in a normal instruction with placeholder. Nevertheless, the instructions still contain some description of the task, and it is easy to imagine that both human and model would make guesses based on the available, incomplete information. The observation that the best language model achieve similar performance to human on such ambiguous task is not surprising, even though the setting is synthetic and simple. In my opinion, I don’t see many new insights from this paper that would help the community to better resolve task ambiguity.


**Summary Of The Paper:**

This paper studies task ambiguity in NLP. Recent research has demonstrated the benefits of using natural language prompts to describe the task when fine-tuning a language model, but such prompts can be ambiguous in some cases. This paper proposes a benchmark with ambiguous instructions, and shows large language models can approach human accuracy on ambiguous tasks. Fine-tuning with a small number of in-context examples further improve the performance.


**Summary Of The Review:**

This paper studies ambiguity in NLP tasks. Although it is an important challenge in fine-tuning language models, this paper only studies a few toy tasks and the proposed method is not novel.

---

> ### Author Response · Authors · 2022-11-14
> **Response [1/2]**
>
> We thank the reviewer for the review! We are happy they find our work addresses an "important challenge that may limit the utility of these methods in the real world."
>
> **"This paper only studies the ambiguity of a few toy tasks, which only require simple lexical understanding and the texts are synthetic and very short. It would be more interesting to see what the ambiguity is in real-world NLP tasks, and how they can be handled."**
>
> We thank the reviewer for the comment and suggestion. It is true that the AmbiBench tasks are simple and synthetic—as an early exploration of task ambiguity in language models we believed a cleaner and more controlled set of experiments would enable a firmer foundation for future work.
>
> That said, we also believe our findings have relevance to real-world settings. To that end, we have added an initial exploration to our paper (Appendix B) expanding our investigation to a more realistic kind of task ambiguity. Specifically, we consider an ambiguously-specified task of parsing an utterance into a command line invocation. Here is an example prompt:
>
> ```
> Write the AWS CLI command to create an AWS Bucket.
>
> Input: Create a bucket for Sato Tamotsu
> Output: aws s3 mb s3://bucket-for-sato --region ap-northeast-1
>
> Input: Create a bucket for Yuki Hashimoto
> Output: aws s3 mb s3://bucket-for-yuki --region ap-northeast-1
>
> Input: Create a bucket for Margaret Richards
> Output:
> ```
>
> Unlike the two-feature ambiguity we consider in AmbiBench, here the ambiguity comes from whether the system should continue the pattern, creating the AWS bucket in the ap-northeast-1 region (Japan), or make the assumption based on the name Margaret Richards that the bucket should be placed in another region (e.g. in Europe or the USA).
>
> We run some initial experiments in this setting, demonstrating that indeed language models behave differently in terms of which hypothesis they favor based on the national origin of the name. We also describe how similar meta-learning via fine-tuning experiments as we explore could teach models the desirable generalization rule, such as "when in doubt, don't make assumptions based on people's names."
>
> We thank the reviewer for this suggestion, as we believe it strengthens the paper by making clear how the cleaner synthetic experiments we explore could be extended into more real-world examples of ambiguity.
>
>
> **The observation that the best language model achieve similar performance to human on such ambiguous task is not surprising, even though the setting is synthetic and simple.**
>
> It is true that the setting is simple, however we believe it is striking how even such simple cases of ambiguity reveal significant differences across models and human behavior. We see performance on Ambibench ranging from far below chance (T0++) to slightly above chance (text-curie-001) to much better than chance (text-davinci-002 and our meta-trained models). We believe this diversity in performance reveals that even a seemingly simple task admits interesting complexity in how the models handle ambiguity.
>
> Furthermore, we computed the performance of an ideal Bayesian oracle that performs inference with respect to a hypothesis class containing both features (Figures 3 and 4, grey dotted line). All models (and humans) are much worse than this oracle, suggesting ample room for future experimentation even in this simple setting.
>
> **The "uninformative instructions" are very unnatural as they just replace the salient feature in a normal instruction with placeholder.**
>
> We chose the placeholder format after some early feedback from human pilot studies who believed something was wrong with the test when they saw more realistic kinds of ambiguity. However, we have conducted some experiments with another kind of uninformative instruction, using the linguistic nonce word "wug" (e.g. as might occur if the user is referencing a person or products that was not present in the model's training data), demonstrating identical behavior. We have updated the paper with these results in Appendix D.
>
> **The novelty is limited. The fine-tuning methods are standard practice in this domain.**
>
> Fine-tuning is certainly not a novel technique, but in this paper we explore fine-tuning as a way to implement *meta-learning* for in-context examples. This technique was only developed earlier this year [1, 2]. Our contribution is to show that the naive form of this technique, training on unambiguous in-context examples, actually yields no benefit. However, if one trains on *ambiguous* in-context examples, this improves a model's ability to learn well via few-shot learning, even on *unseen* ambiguous tasks, so much so that it outperforms the best models trained with reinforcement learning from human feedback.
>
> [1] Yanda Chen, Ruiqi Zhong, Sheng Zha, George Karypis, and He He. Meta-Learning via Language Model In-Context Tuning.
> [2] Sewon Min, Mike Lewis, Luke Zettlemoyer, and Hannaneh Hajishirzi. MetaICL: Learning to Learn in Context

---

> > ### Comment · Reviewer_zof8 · 2022-11-20
> > **Reply to Author Response**
> >
> > Thank the authors for the reply. The following is my assessment for this paper:
> > 1. For the benchmark, the proposed datasets only concern a few toy classification tasks with very short sentences and synthetic instructions. Given that pre-trained language models have nowadays been used in many complex NLP tasks, the contribution of introducing this dataset is limited. The new task added during rebuttal is better, but still it just covers a small area in NLP.
> > 2. For the empirical analysis, there is some merit in this paper for the observation that different models behave differently on these ambiguous tasks. However, I don't find the observation very surprising and inspiring. The paper could elaborate on how this observation can help us to better resolve task ambiguity, or design models based on such inspiration.
> > 3. For the method, in-context learning + finetuning is a pretty straightforward method. As mentioned by the reviewer, it has also been well-studied by other works.
> >
> > In my opinion, it would be much more important to design methods that can deal with the scenarios when a model receives ambiguous questions, such as asking clarification questions, rather than trying to improve the performance on tasks that are inherently ambiguous. So, I would like to retain my score.

---

> ### Author Response · Authors · 2022-11-14
> **Response [2/2]**
>
> **The reproducibility is unclear, as the authors do not mention whether the benchmark will be released.**
> We thank the reviewer for catching this omission—we have updated the paper to make clear that the benchmark and code will be released. ("Reproducibility Statement," page 10)

---

### Official Review · Reviewer_3BBz · 2022-10-28

**Confidence:** 4
**Correctness:** 4
**Technical Novelty And Significance:** 3
**Empirical Novelty And Significance:** 3
**Recommendation:** 6

**Clarity, Quality, Novelty And Reproducibility:**

#### Clarity
- This paper is well written and easy to follow. The experiments are well justified and explained.

#### Quality
- The experiments are well designed but limited in scope.
- The paper would be more valuable if it could could better describe how its findings may extend to broader types of task ambiguity, likely to be seen in downstream tasks of interest.

#### Novelty
- This is the first paper that I've seen that properly investigates task ambiguity of this type in large language models.

#### Reproducability
- The novel work in this paper should be reproducable.
- However, the paper relies heavily on opaque models, and it is a little hard to draw meaningful conclusions about the best performing systems.


**Strength And Weaknesses:**

#### Strengths:
- This is a cleanly designed investigation of a specific type of task ambiguity, which adds to our understanding of how large language models make use of prompts and examples.
- The inclusion of human annotators, assigned the same tasks as the LLMs, really helps us understand the models' capabilities.
- The finetuning results are interesting, and suggest new strategies for increasing the robsutness of LLMs to non-expert prompt design.

#### Weaknesses
- It is quite hard to draw many conclusions about the best performing systems because their training strategies are not public, and it is not clear what types of task ambiguity have been seen during training.
- The six synthetic tasks are quite simplistic and limited in scope. It is not clear how these findings would extend to more complex task types.

#### Minor issues:
- The choices of line color and hatching in Figure 3 are not very discriminative, and I struggled to untangle which line represents which model. I suggest that more disctriminative line types are used.


**Summary Of The Paper:**

This paper focuses on the ability of large language models (LLMs) and humans to handle ambiguity in task prompts, of the type that are currently popular in the LLM literature.

A small suite of six synthetic tasks are created. Each asks the model (or human) to identify whether a sentence contains a particular category (e.g. human subject, proper noun). Then both human and language models are tested in two settings: with the task fully defined in a prompt (e.g. 'output x if the sentence contains a proper noun, y otherwise'); and with the category of interest obfuscated in the prompt. Each task is designed such that there should be a 50% chance of correctly guessing the task type from the obfuscated prompt and a single example. This chance increases drastically as more examples are added.

The paper shows that humans are able to solve five out of six tasks perfectly, when the prompt is unambigious. The performances of LLMs are more varied, with an instruct tuned model, trained with reinforcement-learning from humans (RLFH), performing best but still signifcantly worse than human performance. When the prompt is ambiguous, human performance is closer to model performance. However, when an increasing number of examples are given, the best models are able to disambiguate the taksk much better, reaching >80% accuracy and outperforming humans, in the best case (which involves reading 20 examples). Similar performance is also attainable through fine tuning on 4/6 of the tasks and evaluating on the final 2. This suggests that tuning models to handle task ambiguity may result in generalization to new forms of test time ambiguity.

**Summary Of The Review:**

This paper presents a well designed focused study of how LLMs handle one type of task ambiguity---identifying whether an unknown class of word exists in an input.

A number of different LLMs are evaluated in ambiguous and unambiguous prompting settings, and compared to humans in the same scenarios. The results provide a nice insight into the effectiveness of examples in the context of ambiguous prompts, as well as the effectiveness of fine tuning.

However, the current study focuses on a very narrow type of ambiguity, and it's not clear how much these insights will extend to other types of task ambiguity. This paper would be stronger if it could describe how this type of study could be extended to more diverse types of ambiguity, even if it did not directly address those types of ambiguity in the study presented.

---

> ### Author Response · Authors · 2022-11-14
> **Response**
>
> We thank the reviewer for their review! We are glad they appreciated the clean design of our experiment, the inclusion of human experiments, and the novelty of the questions we investigate.
>
> **It is quite hard to draw many conclusions about the best performing systems because their training strategies are not public, and it is not clear what types of task ambiguity have been seen during training**
>
> We agree! We recognize there is an ongoing discussion in the NLP community about how to respond to blackbox systems whose training details are not public. In light of this, we aimed to strike a balance here in presenting many systems whose training strategies have been made public, as well as the best performing systems overall to give a sense of what is possible. Thus parties can make comparisons with their preferred reference point. (We also note that our benchmark was designed anew for this project, meaning that it could not have been part of the training or RLHF data of any system we evaluate.)
>
> **"This paper would be stronger if it could describe how this type of study could be extended to more diverse types of ambiguity, even if it did not directly address those types of ambiguity in the study presented."**
>
> We thank the reviewer for this suggestion. We have added a discussion to the paper in Appendix B about how one might expand our studies to a more realistic kind of task ambiguity. Specifically, we consider an ambiguously-specified task of parsing an utterance into a command line invocation. Here is an example prompt:
>
> ```
> Write the AWS CLI command to create an AWS Bucket.
>
> Input: Create a bucket for Sato Tamotsu
> Output: aws s3 mb s3://bucket-for-sato --region ap-northeast-1
>
> Input: Create a bucket for Yuki Hashimoto
> Output: aws s3 mb s3://bucket-for-yuki --region ap-northeast-1
>
> Input: Create a bucket for Margaret Richards
> Output:
> ```
>
> Unlike the two-feature ambiguity we consider in AmbiBench, here the ambiguity comes from whether the system should continue the pattern, creating the AWS bucket in the ap-northeast-1 region (Japan), or make the assumption based on the name Margaret Richards that the bucket should be placed in another region (e.g. in Europe or the USA).
>
> We ran some initial experiments in this setting, demonstrating that indeed language models behave differently in terms of which hypothesis they favor. We also note how similar meta-learning via fine-tuning experiments as we explore could teach models the desirable generalization rule, such as "when in doubt, don't make assumptions based on people's names."
>
> We thank the reviewer for this suggestion, as we believe it strengthens the paper by making clear how the cleaner synthetic experiments we explore could be extended into more real-world examples of ambiguity.
>
>
> **The choices of line color and hatching in Figure 3 are not very discriminative, and I struggled to untangle which line represents which model. I suggest that more discriminative line types are used.**
> Thank you for the suggestion. We have made some changes to this figure to make it easier to read in the newest revision.

---

### Official Review · Reviewer_6dKC · 2022-11-28

**Confidence:** 4
**Correctness:** 4
**Technical Novelty And Significance:** 3
**Empirical Novelty And Significance:** 3
**Recommendation:** 6

**Clarity, Quality, Novelty And Reproducibility:**

I am reading the updated version of the paper, which I find to be clearly written and thorough. The experiments are well done with extensive comparisons and a human baseline.

The results are straightforward to reproduce. It would be great to see more experiments with open RLHF models, but there unfortunately are not any available yet.

The synthetic dataset is novel, if simple. This is the first paper to do a clear and self-contained benchmark on task ambiguity (afaict). It evaluates a few known methods for getting the models to accomplish a task (few-shot prompting vs. finetuning) and confirms existing evidence that finetuning performs better.



**Strength And Weaknesses:**

Strengths:
- Simple benchmark with extensive comparisons across model scales and with humans.
  - comparisons to humans grounded the results
- easy to reproduce (even easier assuming the benchmark will be released with camera ready)
- demonstrates a current weakness of LMs in a purely synthetic setting which allow variation.
- preliminary investigation of more real-world tasks, specifically an LM taking actions at command line based on human instructions
- extensive auxiliary experiments in appendix, studying smaller models, alternative prompting choices,

Weaknesses:
- The tasks may be too simple / short / toy.
- Results are not particularly surprising
- Downside of using closed models (although no open models exist for this yet): we do not actually know the training procedure for the largest models.
  - Recent reports suggest that the instruct models are not actually RLHF: https://www.alignmentforum.org/posts/mbGjzyy6eJXT4gFpm/update-to-mysteries-of-mode-collapse-text-davinci-002-not
  - With this in mind, I worry about drawing general conclusions about RLHF
- Small details
  - page 2: what makes finetuning on in-context examples metalearning?
  - page 15: "could could verbalize"

Misc:
- Potential improvement: run additional experiments on the FLAN-T5 models to see how instruction tuning helps smaller models (paper released after submission, so not required but would be useful additional evidence!)




**Summary Of The Paper:**

The paper studies how modern LLMs perform when given ambiguous task examples.

The model must learn to infer the task from the examples, vs having a clear task with ambiguous inputs. The paper introduces a dataset, AmbiBench, designed to measure how well models use the instrutions/few-shot examples to infer underlying task.

AmbiBench consists of 6 sentence classification tasks of minimal complexity which allow controlling and measuring the amount of ambiguity in the tasks. The tasks are all binary classification tasks. Determining the correct underlying task requires either (a) an auxiliary instruction, or (b) multiple examples to learn from.  The examples are generated from a set of templates.

Experiments are carried out with 2 prompt templates across all tasks, and compare a variety of open and closed LLMs, along with human evaluations.

They find:
- for uninformative instructions, both models & humans get 50% accuracy, but humans also get 100% on negation while models do no
- for informative instructions, humans get perfect performance across all tasks but 1, with RLHF models (text-davinci-002) close behind.
- Non-RLHF models do not do nearly as well: davinci does much worse than davinci instruct.
- They further find that ambiguously-specified examples provide a much bigger boost to generalization, when compared to finetuning on unambiguous examples. This provides evidence that finetuning works more robustly than few-shot prompting, and that explicitly training to handle ambiguity helps generalization performance.
- Best model is not able to consistently verbalize the task it is trying to accomplish


Main takeaways from paper: Scaled RLHF models perform best at disambiguating tasks. Finetuning helps over just few-shot prompting

**Summary Of The Review:**

The paper presents a simple synthetic task which measures how many different LMs perform given ambiguous tasks. The synthetic benchmark allows testing this ability extensively and the community will benefit from it if evaluations are easy to run, although its simplicity may make it difficult to generalize the findings.

---

> ### Author Response · Authors · 2022-12-09
> **Response**
>
> We thank the reviewer for the review! We are glad the reviewer appreciates the novelty of the benchmark as well as the extensiveness of our experiments.
>
> **Characterization of models and RLHF**
>
> It appears we can't update our paper at the moment, but in our newest version we characterize these models as being trained with large-scale human feedback data (HFD), rather than RLHF specifically, given the recent documentation OpenAI has provided [1].
>
> **Additional experiments on new models**
>
> We agree the new FLAN-T5 model would be interesting to evaluate, and will benchmark it as well as the newest text-davinci-003 model in the next version of the paper.
>
> **Surprising aspects of the results**
>
> One aspect we found surprising was that, despite the name, the "instruct" models outperformed the human participants *not* on the informative instructions (Section 4.2)  but instead on the multi-example experiments (Section 4.3). This suggests that training with human feedback data may alter a broad range of model characteristics besides instruction following, sometimes in quite unpredictable ways.
>
> It was also pretty surprising to us that meta-learning (finetuning) with just two different pairs of competing features could improve the model's generalization enough to succeed on *held out* ambiguous features. This suggests that altering more general aspects of learning and reasoning in models may be easier to do than commonly thought.
>
> **Typos** Thanks for these—fixed!
>
> We thank the reviewer for again the great questions and suggestions. Please let us know of any remaining concerns.
>
> [1] https://beta.openai.com/docs/model-index-for-researchers

---

### Author Response · Authors · 2022-11-14
**General Response**

We thank the reviewers for their reviews! We have responded to the main comments beneath each review, and also uploaded a new version incorporating these changes. Please do not hesitate to comment with additional questions—we are happy to address any remaining concerns.

---

### Decision · Program_Chairs · 2023-01-20

**Decision:**

Accept: poster

**Justification For Why Not Higher Score:**

Reviewers all noted some weaknesses in the paper, such as the limited nature of the benchmark tasks and the lack of direct application in real-world settings. As such, it’s a promising line of work with clear contributions, but not quite up to the level of warranting an oral presentation.

**Justification For Why Not Lower Score:**

Only one reviewer recommended rejection, on the basis of the tasks being too toy and not related to the challenges of real-world task ambiguity. Other reviewers appreciated the overall approach, execution, and presentation of this work and thought its contributions warranted acceptance. I tend to agree that we need more studies along these lines, ie understanding LLMs using carefully designed and well-controlled tasks.

**Metareview: Summary, Strengths And Weaknesses:**

As this was a borderline paper, with wildly diverging scores (ranging from 3 to 8), a discussion was arranged on Nov. 28 and attended by all reviewers and myself. Reviewers were asked to summarize the main points of their reviews and to consider whether they wanted to maintain their scores in light of seeing other reviews and authors’ rebuttal.

All reviewers agreed that this is a very interesting study, and presented a useful demonstration of how LLMs can be interrogated in a well-controlled setting, in the context of task ambiguity. There seemed to be agreement that, taken individually, the different components of this work weren’t themselves very novel (ie the method itself, the experimental results, or studying task ambiguity), but there was a general sentiment that, all taken together, it was an enlightening study that could prove to be quite impactful in leading to further work along these lines. Further strengths include the inclusion of human experiments on the same task, the release of the AmbiBench benchmark, and the clarity of the execution and presentation. It was noted however that AmbiBench is quite small, synthetic, and toy, although these aspects also allow for well-controlled experiments and better insight into why current large models fail at these tasks, which reviewers appreciated.

One reviewer (zof8) continued to maintain that the tasks used were too toy, and that the more “applied” task added during rebuttals didn’t address the crux of the problem with task ambiguity in the real world - which is that models need to learn how to proactively disambiguate (e.g. by asking clarifying questions), rather than dealing with instructions which are inherently ambiguous. They therefore felt that these tasks are not structured in a way that would generalize to real-world situations.
This reviewer stood by their reject recommendation on these grounds.

Ultimately I tend to agree with the majority of the four reviewers that there is merit to working with well-designed, albeit toy, synthetic tasks, and the insights derived from such experiments constitute important contributions in their own right. I do encourage the authors to look into expanding the set of tasks included in their benchmark, however, as all reviewers noted that they were quite limited in variety. Nonetheless, this work will be of interest to those in the growing field of fine tuning or prompt engineering for pretrained LLMs, and could inform future work in this fast-growing field. I therefore recommend acceptance.


**Note From Pc:**

if the above contains the word "oral" or "spotlight" please see: "oral" presentation means -> notable-top-5% and "spotlight" means -> notable-top-25%. As stated in our emails, we are disassociating presentation type from AC recommendations

**Summary Of Ac-Reviewer Meeting:**

o1Uh: rating = 8 (up from 6)
* This paper is more of a 7, not an 8
* Helps us to understand how pretrained LLMs work
* Tasks are simple but controllable, designed to study ambiguity in an explicit way
* We can clearly see that sota can’t solve this task
* Strength is inclusion of human experiments on the exact same task, which is rather rare
* Not clear how to generalize to other settings
* AmbiBench is more of a tool rather than a dataset release, not a dataset paper
* Insights are perhaps not that deep

zof8: rating = 3 (stayed at 3)
* Agree not a dataset paper, task is too toy, sentence is just a few words
* Perhaps the way that they framed their ambiguity task is not really applicable to real-world ambiguity
* Results aren’t surprising, ie that fine tuning helps with doing this task
* The additional task is helpful, but doesn’t mitigate the other issues
* Method itself is not novel (ie fine tuning, etc)

3BBz: rating = 6
* Perhaps toy tasks are necessary in order to start investigating these questions
* It’s nice that you can boil this task of disambiguation down into binary classification
* The contribution is not in that it’s a benchmark, there need to be more variety of tasks
* However this is quite novel work, not the results necessarily but in the framework itself and how they’re investigating

6dKC: rating = 6
* Pro synthetic and toy datasets (although would be nice to have had more variety of them)
* The method itself isn’t super novel
* Biggest weakness is the closed nature of the models: text-davinci might not actually be trained with RLHF, not clear how these models -are trained, so perhaps we might have a bit less confidence in the generality of the conclusions
* There are no open-sourced models that are RLHF, so not their fault
* Would like to see how larger models do on this task
* EMNLP might be a better fit

Questions:
* Is the new real-world application task enough to make the benchmarks no longer “toy”?
* Who would this paper be for?
* Prompt tuning or prompt engineering community, and anyone else who wants to understand how LLMs work

Follow up discussions
* 3BBz: want to see the follow up paper, could lead to interesting new line of work. Could see this paper being cited, inspire new work
* 6dKC: there aren’t benchmarks like this
* Where exactly is the novelty? 6dKC: Having a setting where you can exactly control the settings, and investigate ambiguity in LLMs. Well-executed.
* zof8: Synthetic tasks are fine, but this particular task isn’t representative of ambiguity in real-world settings, or in NLP. It’s more important to solve problem of how to deal with ambiguity (asking clarifying questions) vs dealing with instructions that are inherently ambiguous. These tasks are not structured in a way that would generalize to real-world situations. Feels quite strongly about his recommendation of reject.
* o1Uh: would make less sense to investigate more complicated tasks before first understanding these synthetic ones. Touched a lot on really interesting topics
* 3BBz: Incredibly hard to control for how prompts are ambiguous, or instructions are ambiguous, so there is merit in having a task that does manage to control for these different aspects. If this paper had more tasks, or the benchmark was more comprehensive, this would be a clear accept.